# Imitating from auxiliary imperfect demonstrations via Adversarial Density Weighted Regression

## Abstract

We propose a novel one-step supervised imitation learning (IL) framework called Adversarial Density Regression (ADR). This IL framework aims to correct the policy learned on unknown-quality to match the expert distribution by utilizing demonstrations, without relying on the Bellman operator. Specifically, ADR addresses several limitations in previous IL algorithms: First, most IL algorithms are based on the Bellman operator, which inevitably suffer from cumulative offsets from sub-optimal rewards during multi-step update processes. Additionally, off-policy training frameworks suffer from Out-of-Distribution (OOD) state-actions. Second, while conservative terms help solve the OOD issue, balancing the conservative term is difficult. To address these limitations, we fully integrate a one-step density-weighted Behavioral Cloning (BC) objective for IL with auxiliary imperfect demonstration. Theoretically, we demonstrate that this adaptation can effectively correct the distribution of policies trained on unknown-quality datasets to align with the expert policy's distribution. Moreover, the difference between the empirical and the optimal value function is proportional to the upper bound of ADR's objective, indicating that minimizing ADR's objective is akin to approaching the optimal value. Experimentally, we validated the performance of ADR by conducting extensive evaluations. Specifically, ADR outperforms all of the selected IL algorithms on tasks from the Gym-Mujoco domain. Meanwhile, it achieves an **89.5%** improvement over IQL when utilizing ground truth rewards on tasks from the Adroit and Kitchen domains.

## 1 Introduction

Reinforcement Learning (RL) has revolutionized various fields, including robotics learning (Brohan et al., 2023a;b; Bhargava et al., 2020), language modeling (Ouyang et al., 2022; Touvron et al., 2023), and the natural science (Gómez-Bombarelli et al., 2018). Despite its success, RL requires extensive interactions with the environment to obtain the optimal policy, which poses challenges for sample efficiency. One way to address this limitation is by leveraging static RL datasets in offline settings. However, this approach often faces the issue of overestimation of Out-Of-Distribution (OOD) states-actions (Levine et al., 2020). To mitigate this, prior research has introduced conservative methods, such as incorporating regularization terms (Fujimoto et al., 2019a; Wu et al., 2022) in the policy learning objective, or pessimism terms in value function learning objective (Kumar et al., 2020a), helping alleviate the OOD issues. However, offline RL algorithms generally assume that offline datasets contain reward labels. Moreover, striking the balance with conservative terms in offline RL remains difficult, particularly for tasks with sparse rewards (Cen et al., 2024).

On the other hand, when the dataset does not contain rewards, we can utilize Imitation Learning (IL) algorithms to learn near-expert policy by utilizing a large amount of unknown-quality datasets and a small number of demonstrations (Argall et al., 2009). In particular, one of the most common methods is to train a discriminator through generative Adversarial Learning to represent the reward or value functions (Ho and Ermon, 2016), and followed by updating within RL frameworks. However, it is difficult for the discriminator to converge to its optimal value (Kostrikov et al., 2019). Furthermore, sub-optimal reward or value functions can lead to unstable training. On the other hand, there is another approach termed distribution correct estimation (DICE) (Kim et al., 2022; Ma et al., 2022a;

Reddy et al., 2019). It corrects the policy's distribution through importance sampling (IS) to make the learned policy closer to the expert's distribution. However, the cumulative offset caused by suboptimal rewards or values in the process of using the Bellman operator for multi-step updates has not been fundamentally resolved, and balancing conservatism remains challenging.

To address these limitations, we introduce ADR, a streamlined one-step supervised framework derived from Equation 7. The key objective of ADR is to closely align the policy distribution with that of the demonstrations while diverging from the distributions of datasets with unknown-quality. Theoretically, this method effectively shifts the empirical distribution toward the expert distribution in a direct and corrective manner (Proposition 5.2). Moreover, we demonstrate in Proposition 5.3 that the value bound is proportional to the lower bound of ADR's objective. Thus, minimizing ADR's objective leads to convergence towards the optimal policy. In particular, ADR is a one-step supervised IL framework, where all training samples are in-sample, effectively eliminating the challenges of OOD issues. This approach is particularly promising in offline settings, as most RL studies frame the offline RL problem within a Markov Decision Process (MDP) (Kumar et al., 2019; Kostrikov et al., 2021; Haarnoja et al., 2018a; Fujimoto et al., 2019b; van Hasselt et al., 2015). Under the MDP setting, decision-making depends solely on the current observation and policy, independent of historical information. Thus, if the action support is adequately relocated, the policy's performance can be ensured. To validate ADR's effectiveness, we evaluated it across various tasks from the Adroit and Gym-Mujoco domains under the Learning from Demonstration (LfD) setting, where it demonstrated competitive results. Notably, ADR outperformed Implicit Q Learning (IQL) by 89.5% on tasks from the Adroit and Kitchen domains when utilizing ground truth rewards.

Our main contribution is ADR, a novel single-step supervised IL method. Unlike most modern RL-combined IL algorithms, which rely on the Bellman operator and incorporate reward shaping and Q-estimating processes, ADR operates as a single-step supervised learning paradigm, rendering it immune to the accumulated offsets resulting from suboptimal rewards. Meanwhile, ADR neither requires the addition of conservative terms nor extensive hyperparameter parameter tuning during the training process. Meanwhile, compared to traditional single-step IL paradigms such as Behavioral Cloning (BC), ADR can achieve better performance with a limited number of demos based on adversarial density-weighted regression. Therefore, ADR combines the advantages of single-step updates while demonstrating superior performance compared to previous RL-combined IL approaches on the experimental level. Moreover, we prove that optimizing ADR's objective is akin to approaching the demo policy, and our experimental results validate this claim, demonstrating that ADR outperforms the majority of RL-combined approaches across diverse domains.

## 2 RELATED WORK

**Imitation Learning (IL).** IL has a long history of development, with well-known algorithms such as BC. However, BC is brittle when demonstrations are scarcity (Ross et al., 2011a). Currently, the more effective IL paradigms are generally of the RL-combined type. Specifically, these type of IL methods encompass various settings, each tailored to specific objectives. Primarily, IL can be categorized based on the imitating objective into Learning from Demonstration (LfD) (Argall et al., 2009; Judah et al., 2014; Ho and Ermon, 2016; Brown et al., 2020; Ravichandar et al., 2020; Boborzi et al., 2022) and Learning from Observation (LfO) (Ross et al., 2011b; Liu et al., 2018; Torabi et al., 2019; Boborzi et al., 2022). Despite RL-combined RL methods have shown improved performance, most RL-combined IL algorithms are based on reward or Q-value estimation. Therefore, this paradigm may suffer from cumulative offsets originating from suboptimal rewards, which can affect the performance of the policy. To overcome this limitation, we introduce ADR that utilizes a density-weighted BC objective to perform single-step updates, effectively mitigating cumulative offsets while preserving high performance as RL-combined methods. Additionally, IL can also be implemented in a supervised learning manner by training a latent information-conditioned policy (Liu et al., 2023a; Zhang et al., 2024). However, they introduce an extra latent condition.

**Behavior Policy Modeling.** Previously, estimating the support of the behavior policy has been approached using various methods, including Gaussian (Kumar et al., 2019; Wu et al., 2019) or Gaussian mixture (Kostrikov et al., 2021) sampling approaches, Variance Auto-Encoder (VAE) based techniques (Kingma and Welling, 2022; Debbagh, 2023), or accurate sampling via auto-regressive language models (Germain et al., 2015). Specifically, the most relevant research to our

study involves utilizing VAE to estimate the density-based definition of action support (behavior density) (Fujimoto et al., 2019b; Wu et al., 2022). On the other hand, behavior policy is utilized to regularize the offline training policy (Fujimoto and Gu, 2021), reducing the extrapolation error of offline RL algorithms, it has also been utilized in offline-to-online setting (Wu et al., 2022; Fujimoto and Gu, 2021; Nair et al., 2021) to ensure the stable online fine-tuning. Different from the previous study, our focus is on using the estimated target density to optimize policy with the ADR objective.

## 3 PRELIMINARIES

**Reinforcement Learning (RL).** We consider RL can be represented by a Markov Decision Process (MDP) tuple *i.e.*, $\mathcal{M} := (\mathcal{S}, \mathcal{A}, p_0, r, d_{\mathcal{M}}, \gamma)$, where $\mathcal{S}$ and $\mathcal{A}$ separately denotes observation and action space, $\mathbf{a} \in \mathcal{A}$ and $\mathbf{s} \in \mathcal{S}$ separately denotes state (observation) and action (decision making). $\mathbf{s}_0$ denotes initial observation, $p_0$ denotes initial distribution, $r(\mathbf{s}_t, \mathbf{a}_t) : \mathcal{S} \times \mathcal{A} \to \mathbb{R}$ denotes reward function. $d_{\mathcal{M}}(\mathbf{s}_{t+1}|\mathbf{s}_t, \mathbf{a}_t) : \mathcal{S} \times \mathcal{A} \to \Delta(\mathcal{S})$ denotes the transition function, $\gamma \in [0, 1]$ denotes the discounted factor. The goal of RL is to obtain the optimal policy $\pi^*$ that can maximize the accumulated Return *i.e.*, $\pi^* := \arg\max_\pi \sum_{t=0}^{t=T} \gamma^t \cdot r(\mathbf{s}_t, \mathbf{a}_t)$, where $\tau = \{\mathbf{s}_0, \mathbf{a}_0, r(\mathbf{s}_0, \mathbf{a}_0), \cdots, \mathbf{s}_k, \mathbf{a}_k, r(\mathbf{s}_k, \mathbf{a}_k), \cdots, \mathbf{s}_T, \mathbf{a}_T, r(\mathbf{s}_T, \mathbf{a}_T)|\mathbf{s}_0 \sim p_0, \mathbf{a}_t \sim \pi(\cdot|\mathbf{s}_t), \mathbf{s}_{t+1} \sim d_{\mathcal{M}}(\cdot|\mathbf{s}_t, \mathbf{a}_t)\}$, and $T$ denotes time horizon.

**Imitation Learning (IL).** In IL problem setting, $r(\mathbf{s}, \mathbf{a})$ is inaccessible, but we have access to a limited number of demonstrations $\mathcal{D}^* = \{\tau^* = \{\mathbf{s}_0, \mathbf{a}_0, \mathbf{s}_1, \mathbf{a}_1, \cdots, \mathbf{s}_k, \mathbf{a}_k, \cdots \mathbf{s}_T, \mathbf{a}_T | \mathbf{a}_t \sim \pi^*(\cdot|\mathbf{s}), \mathbf{s}_0 \sim p_0, \mathbf{s}_{t+1} \sim d_{\mathcal{M}}(\mathbf{s}_{t+1}|\mathbf{s}_t, \mathbf{a}_t)\}\}$, and large amount of unknown-quality dataset $\hat{\mathcal{D}} = \{\hat{\tau}|\hat{\tau} \sim \hat{\pi}\}$. In particular, one of the classical IL methods is behavior cloning (BC), where the objective is to maximize the likelihood of expert decision-making, as follows:

$$\pi_\theta := \arg\max_{\pi_\theta} \mathbb{E}_{(\mathbf{s},\mathbf{a})\sim\mathcal{D}^*}[\log \pi_\theta(\mathbf{a}|\mathbf{s})], \tag{1}$$

however, BC's performance is brittle when $\mathcal{D}^*$ is scarcity (Ross et al., 2011a). Another approach is to recover a policy $\pi(\cdot|\mathbf{s})$ by matching the distribution of the expert policy. Since $\pi^*$ cannot be directly accessed, previous studies frame IL as a distribution-matching problem. Specifically, the process begins by estimating a reward or Q-function $c(\mathbf{s}, \mathbf{a})$ as follows:

$$c(\mathbf{s}, \mathbf{a}) := \arg\min_c \mathbb{E}_{(\mathbf{s},\mathbf{a})\sim\hat{\mathcal{D}}}[\log(\sigma(c(\mathbf{s}, \mathbf{a})))] + \mathbb{E}_{(\mathbf{s},\mathbf{a})\sim\mathcal{D}^*}[\log(1 - \sigma(c(\mathbf{s}, \mathbf{a})))], \tag{2}$$

where $\sigma$ denotes the *Sigmoid* function. The empirical policy $\pi_\theta$ is then optimized within a RL framework. However, most of these approaches rely on Adversarial learning (Kostrikov et al., 2019), which often suffers from unstable training caused by sub-optimal reward or value functions.

**Behavior density estimation via Variance Auto-Encoder (VAE).** Typically, action support constrain *i.e.*, $D_{\mathrm{KL}}[\pi_\theta||\pi_\beta] \leq \epsilon$ has been utilized to confine the training policy to the support set of the behavior policy $\pi_\beta$ (Kumar et al., 2019; Fujimoto et al., 2019b), aiming to mitigate extrapolation error. In this research, we propose leveraging existing datasets and demonstrations to separately learn the target and sub-optimal behavior densities, which are then utilized for ADR. In particular, we follow Wu et al. to estimate the density of action support with Linear Variance Auto-Encoder (VAE) (as demonstrated VAE-1 in (Damm et al., 2023)) by Empirical Variational Lower Bound (ELBO) :

$$\log p_\Theta(\mathbf{a}|\mathbf{s}) \geq \mathbb{E}_{q_\Phi(\mathbf{z}|\mathbf{a},\mathbf{s})}[\log p_\Theta(\mathbf{a}, \mathbf{z}|\mathbf{s})] - D_{\mathrm{KL}}[q_\Phi(\mathbf{a}|\mathbf{s}, \mathbf{a})||p(\mathbf{s}|\mathbf{z})]$$
$$\stackrel{\text{def}}{=} -\mathcal{L}_{\mathrm{ELBO}}(\mathbf{s}, \mathbf{a}; \Theta, \Phi), \tag{3}$$

and computing the policy likelihood through importance sampling during evaluation:

$$\log p_\Theta(\mathbf{a}|\mathbf{s}) \approx \mathbb{E}_{\mathbf{z}^l \sim q_\Phi(\mathbf{z}|\mathbf{s},\mathbf{a})}\left[\frac{1}{L}\sum_L \frac{p_\Theta(\mathbf{a}, \mathbf{z}^l|\mathbf{s})}{q_\Phi(\mathbf{z}^l|\mathbf{a}, \mathbf{s})}\right] \stackrel{\text{def}}{=} \mathcal{L}_{\pi_\beta}(\mathbf{s}, \mathbf{a}; \Theta, \Phi, L), \tag{4}$$

where $\mathbf{z}^l \sim q_\Phi(\mathbf{z}|\mathbf{s}, \mathbf{a})$ is the $l_{th}$ sampled VAE embedding, $\Theta$ and $\Phi$ are separately encoder's and decoder's parameter, $l$ and $L$ respectively denote the $l_{th}$ sampling index and the total sampling times.

# 4 PROBLEM FORMULATION

**Notations.** Prior to formulating our objective, we first define $P^*(\mathbf{a}|\mathbf{s})$ as the expert behavior density (*The conception of behavior density is proposed by Wu et al. (2022), representing the density probability of the given action* $\mathbf{a}$ *within the expert action support*), and define the sub-optimal behavior density as $\hat{P}(\mathbf{a}|\mathbf{s})$. Meanwhile, we define the training policy as $\pi_\theta(\cdot|\mathbf{s}) : \mathcal{S} \to \mathcal{A}$. Additionally, we denote the stationary distributions of the empirical policy, datasets and expert policy by $d^\pi$, $d^\mathcal{D}$ and $d^{\pi^*}$, respectively. The Kullback-Leibler (KL) divergence is represented as $D_{\mathrm{KL}}$. where $d^{\pi^*}(\mathbf{s}, \mathbf{a})$ can be formulated by replacing $\pi$ with $\pi^*$.

**Definition 1.** *(Stationary Distribution) We separately define the $\gamma$ discounted stationary distribution (state-action occupancy) of expert and non-expert behavior as $d^{\pi^*}(\mathbf{s}, \mathbf{a})$ and $d^\pi(\mathbf{s}, \mathbf{a})$. In particular, $d^\pi(\mathbf{s}, \mathbf{a})$ can be formulated as:*

$$d^\pi(\mathbf{s}, \mathbf{a}) := (1 - \gamma) \sum_{t=0}^\infty \gamma^t \cdot \mathrm{Pr}(\mathbf{s} = \mathbf{s}_t, \mathbf{a} = \mathbf{a}_t | \mathbf{s}_0 \sim \mu_0, \mathbf{a}_t \sim \pi(\cdot|\mathbf{s}_t), \mathbf{s}_{t+1} \sim d_\mathcal{M}(\cdot|\mathbf{s}_t, \mathbf{a}_t)), \quad (5)$$

Previous IL algorithms have several limitations: 1) Accumulated offsets can result from using sub-optimal reward or value functions during multi-step updates. Additionally, off-policy frameworks may introduce OOD state-actions. 2) Some off-policy offline frameworks necessitate tuning of hyperparameters to strike a balance between conservatism, and overly conservatism constrains the exploratory capacity of policies, limiting their ability to adapt and improve beyond the demonstrations provided. To overcome these issues, we completely adapt a supervised learning objective ADR to correct the policy distribution on unknown-quality datasets using a small number of demonstrations.

**Remark 4.1.** $d^\pi(\mathbf{s}) > 0$ *whenever $d^\mathcal{D}(\mathbf{s}) > 0$ is a guarantee that the on-policy samples $\mathcal{D}$ has coverage over the expert state-marginal, and is necessary for IL to succeed. (This remark has been extensively deliberated by Ma et al.)*

**Policy Distillation via KL Divergence.** Rusu et al. (2016) demonstrates the effectiveness of policy distillation by minimizing the KL divergence between the training policy $\pi_\theta$ and the likelihood of teacher policy set $\pi_i \in \Pi$, *i.e.*, $\pi := \arg\min_{\pi_\theta} D_{\mathrm{KL}}[\pi_\theta || \pi_i]\big|_{\pi_i \in \Pi}$. Meanwhile, if the condition mentioned in Remark 4.1 is held, we can directly achieve expert behavior through distillation, *i.e.*,

$$\pi := \arg\min_{\pi_\theta} D_{\mathrm{KL}}[\pi_\theta || P^*]. \quad (6)$$

however, it's insufficient to mimic the expert behavior by minimizing the KL divergence between $\pi_\theta(\mathbf{a}|\mathbf{s})$ and $P^*(\mathbf{a}|\mathbf{s})$, since the limited demonstrations aren't sufficient to help to estimate a good $P^*(\cdot|\mathbf{s})$. To address this limitation, we propose Adversarial Density Regression (ADR), a supervised learning algorithm that utilizes a limited number of demonstrations to correct the distribution learned by the policy on datasets of unknown-quality, thereby bringing it closer to the expert distribution.

**Adversarial Density Regression (ADR).** In particular, beyond aligning $\pi_\theta$ with the expert distribution $P^*$, we also push $\pi_\theta$ away from the empirical distribution $\hat{P}$, as formulated in Equation 7. This approach is formalized as Adversarial Density Regression (ADR) in Definition 2. The primary advantage of ADR lies in its independence from the Bellman operator, and it's an one-step supervised learning paradigm. Therefore, ADR won't be impacted by the cumu-

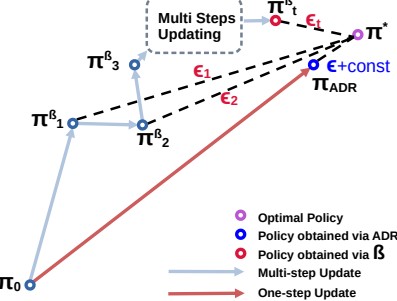

Figure 1: Blue path based on Bellman operator $\mathcal{B}$, the distance from the optimal policy varies with all iterations. Red path, the precise path to the optimal policy.

lative offsets that introduced during multi-step updates (demonstrated in Figure 1), ensuring a more stable and reliable learning process.

**Definition 2** (Adversarial Density Regression (ADR)). *Given expert behavior density $P^*(\mathbf{a}|\mathbf{s})$ and sub-optimal behavior density $\hat{P}(\mathbf{a}|\mathbf{s})$, we formulate the process of Adversarial Policy Divergence,*

*where $\pi_\theta$ approaches the expert behavior while diverging from the sub-optimal behavior, as follows:*

$$\pi_\theta := \underset{\pi_\theta}{\arg\min} \, \mathbb{E}_\mathcal{D}[D_{\mathrm{KL}}[\pi_\theta||P^*] - D_{\mathrm{KL}}[\pi_\theta||\hat{P}]], \tag{7}$$

**Density Weighted Regression (DWR).** However, it's computing in-efficient to directly compute the objective formulated in Definition 2. But, according to Theorem 4.2, we can instead computing:

$$\pi_\theta := \underset{\pi_\theta}{\arg\min} \, \mathbb{E}_{(\mathbf{s},\mathbf{a})\sim\mathcal{D}} \big[\mathcal{W}(\hat{P}, P^*) \cdot ||\pi_\theta(\cdot|\mathbf{s}) - \mathbf{a}||^2\big] \tag{8}$$

to replace Equation 7, where $\mathcal{W}(\hat{P}, P^*) = \log \frac{\hat{P}(\mathbf{a}|\mathbf{s})}{P^*(\mathbf{a}|\mathbf{s})}\big|_{(\mathbf{s},\mathbf{a})\sim\mathcal{D}}$ termed **density weight**.

**Theorem 4.2** (Density Weight). *Given expert log behavior density $\log P^*(\mathbf{a}|\mathbf{s}) : \mathcal{S} \times \mathcal{A} \to \mathbb{R}$, sub-optimal log behavior density $\log \hat{P}(\mathbf{a}|\mathbf{s}) : \mathcal{S} \times \mathcal{A} \to \mathbb{R}$, and the empirical policy $\pi_\theta : \mathcal{S} \to \mathcal{A}$, offline dataset $\mathcal{D}$. Minimizing the KL divergence between $\pi_\theta(\mathbf{a}|\mathbf{s})$ and $P^*(\mathbf{a}|\mathbf{s})$, while maximizing the KL divergence between $\pi_\theta(\mathbf{a}|\mathbf{s})$ and $\hat{P}(\mathbf{a}|\mathbf{s})$, i.e.Equation 7. is equivalent to:*

$$\pi_\theta := \underset{\pi_\theta}{\arg\min} \, \mathbb{E}_{(\mathbf{s},\mathbf{a})\sim\mathcal{D}} \big[ \log \frac{\hat{P}(\mathbf{a}|\mathbf{s})}{P^*(\mathbf{a}|\mathbf{s})} \cdot ||\pi_\theta(\cdot|\mathbf{s}) - \mathbf{a}||^2\big], \tag{9}$$

*Proof* of Theorem 4.2, see Appendix D.1.

Meanwhile, to further address the limitations of BC's tendency to overestimate given state-action pairs, we propose minimizing the upper bound of Equation 8 during each update epoch. This approach serves as an alternate real optimization objective, mitigating the overestimation issues *i.e.*,

$$\min_{\pi_\theta} J(\pi_\theta) = \min_{\pi_\theta} \mathbb{E}_{\beta_\mathcal{D}\sim\mathcal{D}}\mathbb{E}_{(\mathbf{s},\mathbf{a})\sim\beta_\mathcal{D}} \big[\mathcal{W}(\hat{P}, P^*) \cdot ||\pi_\theta(\cdot|\mathbf{s}) - \mathbf{a}||^2\big] \tag{10}$$

$$(\textbf{Cauchy's Inequality}) \leq \min_{\pi_\theta} \mathbb{E}_{\beta_\mathcal{D}\sim\mathcal{D}}\mathbb{E}_{(\mathbf{s},\mathbf{a})\sim\beta_\mathcal{D}} \big[\mathcal{W}(\hat{P}, P^*)\big] \cdot \mathbb{E}_{(\mathbf{s},\mathbf{a})\sim\beta_\mathcal{D}} [||\pi_\theta(\cdot|\mathbf{s}) - \mathbf{a}||^2], \tag{11}$$

where $\beta_\mathcal{D} \in \mathcal{D}$ denotes a batch sampled offline dataset during the offline training process.

## 5 THEORETICAL ANALYSIS OF ADVERSARIAL DENSITY REGRESSION

In this section, we further conduct a theoretical analysis to demonstrate the convergence of ADR.

**Assumption 5.1.** *Suppose the policy extracted from Equation 11 is $\pi$, we separately define the state marginal of the dataset, empirical policy, and expert policy as $d^\mathcal{D}$, $d^\pi$ and $d^{\pi^*}$, they satisfy this relationship:*

$$D_{KL}[d^\pi||d^{\pi^*}] \leq D_{KL}[d^\mathcal{D}||d^{\pi^*}] \tag{12}$$

**Proposition 5.2** (Policy Convergence of ADR). *Assuming Equation 7 can finally converge to $\epsilon$ via minimizing Equation 9, meanwhile, assuming Assumption D.2 is held. Then $\mathbb{E}_{(\mathbf{s},\mathbf{a})\sim\hat{\mathcal{D}}}[D_{KL}(\pi||\pi^*)] \to \frac{M}{2n} \cdot \sqrt{\log\frac{2}{\delta}} + \Delta C + \epsilon$.*

**Proposition 5.3.** *(Value Bound of ADR) Given the empirical policy $\pi$ and the optimal policy $\pi^*$, let $V^\pi(\rho_0)$ and $V^{\pi^*}(\rho_0)$ separately denote the value network of $\pi$ and $\pi^*$, and given the discount factor $\gamma$. Meanwhile, let $R_{max}$ as the upper bound of the reward function i.e., $R_{max} = \max ||r(\mathbf{s},\mathbf{a})||$. Based on the Assumption D.7, Assumption D.2, Lemma D.8, and Proposition 5.2, we can obtain:*

$$|V^\pi(\rho_0) - V^{\pi^*}(\rho_0)| \leq \underbrace{\frac{R_{max}}{1-\gamma} D_{TV}[d^*(\mathbf{s})||d^\mathcal{D}(\mathbf{s})]}_{w.l.o.g} + \frac{2 \cdot R_{max}}{1-\gamma} \cdot \sqrt{2 \cdot (\frac{M}{2n} \cdot \sqrt{\log\frac{2}{\delta}} + \Delta C + \epsilon)}, \tag{13}$$

*where, $\Delta C = C_1 - C_2$ is a constant term, dependent on the state distribution. $\delta$ originates from Assumption D.2, $n = |\mathcal{D}^*|$, $M := \arg\max_{X_i}\{X_i = \pi^*(\mathbf{a}_t|\mathbf{s}_t)\log\frac{\pi^*(\mathbf{a}_t|\mathbf{s}_t)}{\hat{\pi}(\mathbf{a}_t|\mathbf{s}_t)}|(\mathbf{s}_t,\mathbf{a}_t)\sim\mathcal{D}^*\}$.*

*Proof* of Proposition 5.2 and Proposition 5.3, see Appendix D.4 and Appendix D.9.

From Proposition 5.2, we can infer that if Equation 7 converges to a small threshold $\epsilon$, the KL divergence between the likelihood of $\pi$ and $\pi^*$ on unknown-quality data will converge to the same order of magnitude *i.e.*, $O(\epsilon)$. This implies that the action distribution learned by the $\pi$ will become closer to the $\pi^*$, as long as the states in the unknown-quality data sufficiently cover the states of the $\pi^*$, $\pi$ will learn as many expert decisions as possible. At the same time, in Proposition 5.3, we further prove that the regret of policy $\pi$ is proportional to the convergence upper bound of Equation 7. Therefore, minimizing Equation 7 implies that $V^\pi(\rho_0)$ will converge to the $V^{\pi^*}(\rho_0)$ considering the current dataset. Specifically, the first term on the left-hand side of Equation 13 is determined by the quality of the dataset, which is generally applicable to all algorithms (**w.l.o.g**). However, the second term is unique to ADR, as the supervised optimization objective of ADR aligns with maximizing $V^\pi(\rho_0)$. Therefore, minimizing ADR's objective can bring $\pi$ closer to $\pi^*$

**Policy Distribution Analysis.** To validate the near-optimal policy convergence, we visualize the policy distribution of both the behavior learned by ADR and expert behavior (sampled from dataset) in Figure 2. Remarkably, utilizing solely the `medium-replay` dataset, ADR is able to comprehensively cover the expert behavior, demonstrating its efficacy in mimicking the expert policy, thus validaing our claim in Proposition 5.2.

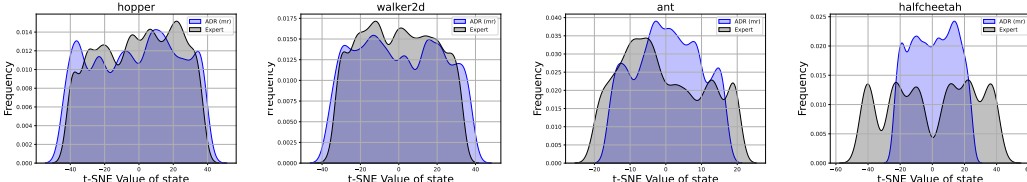

Figure 2: Policy Distribution. We sequentially sampled 500 samples $\tau_{\texttt{sampled}} = \{(\mathbf{s}_t, \mathbf{a}_t) | (\mathbf{s}_t, \mathbf{a}_t) \sim \mathcal{D}_{\exp}\}_{t=0}^{t=500}$ from the expert dataset $\mathcal{D}_{\exp}$. At the same time, we generated 500 actions based on the policy learned from ADR *i.e.*, $\tau_{\texttt{generate}} = \{\mathbf{a}_t := \pi_\theta(\cdot|\mathbf{s}_t) | \mathbf{s}_t \in \tau_{\texttt{sampled}}\}$. Then, we reduced the dimensions of actions from all $\tau_{\texttt{sampled}}$ and $\tau_{\texttt{generate}}$ using t-SNE and plot the KDE curve.

# 6 METHODS

To alleviate the constraint posed by the scarcity of demonstrations, we introduce Adversarial Density Estimation (ADE).

**Adversarial Density Estimation (ADE).** Specifically, during the training stage, we utilize the ELBO of VAE to estimate the density probability of state-action pair in action support *i.e.*, Equation 4. Additionally, to alleviate the limitation of demonstrations' scarcity, we utilize adversarial learning (AL) in density estimation. This involves maximizing the density probability of expert offline samples while minimizing the density probability of sub-optimal offline samples to improve the estimation of expert behavior density. ($\Theta^*$ *doesn't mean the optimal parameter, instead, it means the parameters of VAE model utilized to estimate on expert samples*) :

$$\mathcal{J}_{\text{ADE}}(\Theta^*) = \mathbb{E}_{(\mathbf{s},\mathbf{a})\sim\pi^*}\left[\sigma(P_{\Theta^*}(\mathbf{a}|\mathbf{s}))\right] - \mathbb{E}_{(\mathbf{s},\mathbf{a})\sim\hat{\pi}}\left[\sigma(P_{\Theta^*}(\mathbf{a}|\mathbf{s}))\right], \tag{14}$$

Therefore, the expert density's objective can be formulated as :

$$\mathcal{J}(\Theta^*) = \mathbb{E}_{(\mathbf{s},\mathbf{a})\sim\pi^*}\left[\mathcal{L}_{\text{ELBO}}(\mathbf{s},\mathbf{a};\Theta^*,\Phi^*)\right] + \lambda \cdot \mathcal{J}_{\text{ADE}}(\Theta^*). \tag{15}$$

Accordingly, the objective for non-expert density can be formulated by substituting $\Theta^*$ and $\Phi^*$ in Equation 15 with $\hat{\Theta}$ and $\hat{\Phi}$. However, in practical implementations, we find that setting $\lambda = 0$ is sufficient to achieve good performance for sub-optimal behavior density.

**Density Weighted Regression (DWR).** After using ADE and obtaining the converged VAE estimators $P_{\Theta^*}(\mathbf{a}|\mathbf{s})$ and $P_{\hat{\Theta}}(\mathbf{a}|\mathbf{s})$. We freeze the parameter of these estimators, then approximate the

**density weight** $W(\hat{P}, P^*) = \log \frac{\hat{P}(\mathbf{a}|\mathbf{s})}{P^*(\mathbf{a}|\mathbf{s})}$ using importance sampling:

$$\log \frac{\hat{P}(\mathbf{a}|\mathbf{s})}{P^*(\mathbf{a}|\mathbf{s})}\bigg|_{(\mathbf{s},\mathbf{a})\sim\mathcal{D}} \approx \log p_{\hat{\Theta}}(\mathbf{a}|\mathbf{s}) - \log p_{\Theta^*}(\mathbf{a}|\mathbf{s})\bigg|_{(\mathbf{s},\mathbf{a})\sim\mathcal{D}} \tag{16}$$
$$\approx \mathcal{L}_{\pi_\beta}(\mathbf{s}, \mathbf{a}; \hat{\Theta}, \hat{\Phi}, L) - \mathcal{L}_{\pi_\beta}(\mathbf{s}, \mathbf{a}; \Theta^*, \Phi^*, L)\big|_{(\mathbf{s},\mathbf{a})\sim\mathcal{D}},$$

and then bring density weight into Equation 10 or 11, optimizing policy via gradient decent *i.e.*, $\theta \leftarrow \theta - \eta \cdot \nabla_\theta \mathcal{J}(\pi_\theta)$, where $\eta$ denotes learning rate (lr).

### 6.1 PRACTICAL IMPLEMENTATION

ADR comprises VAE Pre-training (Algorithm 1) and policy training (Algorithm 2) stages. During the VAE pre-training stage, we utilize VQ-VAE to separately estimate the target density $P^*(\mathbf{a}|\mathbf{s})$ and the suboptimal density $\hat{P}(\mathbf{a}|\mathbf{s})$ by minimizing Equation 11 (or Equation 15) and the VQ loss (van den Oord et al., 2018). During the policy training stage, we optimize the Multiple Layer Perception (MLP) policy $\pi_\theta$ by using Equation 8. For more details about our model architecture and more hyper-parameter settings, please refer to Appendix. In terms of evaluation. We compute the normalized D4rl (normalized) score with the same method as Fu et al., and our experimental result is obtained by averaging the highest score in multiple runs.

---

**Algorithm 1** VAE Pretraining

**Require:** VAE (density estimator) parameterized by $(\Theta^*, \Phi^*)$ for expert dataset, VAE parameterized by $(\hat{\Theta}, \hat{\Phi})$ for unknown-quality dataset. Empirical policy $\pi_\theta(\cdot|\mathbf{s})$, unknown-quality offline datasets $\hat{\mathcal{D}}$, demonstrations $\mathcal{D}^*$; VAE training epochs $N_{VAE\ train}$ and policy training epochs $N_{policy\ train}$.

1: **while** $t_1 \leq N_{VAE\ train}$ **do**
2:     Sample batch sub-optimal trajectory $\hat{\tau}$ from $\hat{\mathcal{D}}$, and sampling batch expert trajectory $\tau^*$ from $\mathcal{D}^*$.
3:     update $(\Theta^*, \Phi^*)$ by Equation 15. Replace $(\Theta^*, \Phi^*)$ in Equation 15 with $(\hat{\Theta}, \hat{\Phi})$, and update $(\hat{\Theta}, \hat{\Phi})$.
4: **end while**

**Algorithm 2** Training Policy

**Require:** pre-trained density estimators $\hat{P}, P^*$, and datasets $\mathcal{D} = \hat{\mathcal{D}} \cup \mathcal{D}^*$

1: **while** $t_2 \leq N_{policy\ train}$ **do**
2:     Computing $\mathcal{W}(\hat{P}, P^*) = \log \frac{P_{\hat{\Phi}}(\mathbf{a}|\mathbf{s})}{P_{\Phi^*}(\mathbf{a}|\mathbf{s})}$.
3:     Bring $\mathcal{W}(\hat{P}, P^*)$ to Equation 11 or 8 and updating $\pi_\theta$.
4: **end while**

---

## 7 EVALUATION

Our experiments are designed to answer: 1) Does ADR outperform prvious IL approaches (include DICE)? 2) Is it necessary to use an adversarial approach to assist in estimating the target behavior density? 3) Is it necessary to use the density-weighted form to optimize the policy?

**Datasets.** The majority of our experimental setups are centered around Learning from Demonstration (LfD). For convenience, we denote using n demonstrations to conduct experiments under the LfD setting as LfD (n). We test our method on various domains, including Gym-Mujoco, Android, and Kitchen domains (Fu et al., 2021). Specifically, the datasets from the Gym-Mujoco domain include `medium` (m), `medium-replay` (mr), and `medium-expert` (me) collected from environments including `Ant`, `Hopper(hop)`, `Walker2d(wal)`, and `HalfCheetah(che)`, and the demonstrations are 5 expert trails from the respective environments. For the `kitchen` and `androits` domains, we rank and sort all trials by their return, and sample the trial with the highest return as demonstration. *The content inside the parentheses* `()` *represents an abbreviation.*

**Baselines.** The majority selected baselines are shown in Table 3. Specifically, when assessing the Gym-Mujoco domain, the baselines encompass ORIL, SQIL, IQ-Learn, ValueDICE, DemoDICE, SMODICE utilized RL-based weighted BC approaches to update. Additionally, we also compared with previous competitive contextualized BC framework CEIL. When test on kitchen or androits domains, we compared our methods with IL algorithms including OTR and CLUE that utilize reward relabeling approach, and policy optimization via Implicit Q Learning (IQL) (Kostrikov et al., 2021), besides, we also compare ADR with Conservative Q Learning (CQL) (Kumar et al., 2020b) and IQL utilizing ground truth reward separately denoted CQL (oracle) and IQL (oracle), where oracle

Table 1: **Previous IL approaches**. We summarize the majority of previous IL approaches here. Specifically, most of these methods involve estimating the reward or value function and are followed by optimizing with the weighted BC objective.

| Algorithm | Optimizing framework | estimating Target | Methods for Target estimating | Weighted BC |
|---|---|---|---|---|
| OTR (Luo et al., 2023) | IQL (Kostrikov et al., 2021) | $r(\mathbf{s}, \mathbf{a})$ | Wasserstein Distance | ✔ |
| SQIL (Reddy et al., 2019) | IQL | $r(\mathbf{s}, \mathbf{a})$ | Const Reward | ✔ |
| CLUE (Liu et al., 2023b) | IQL | $r(\mathbf{s}, \mathbf{a})$ | $L_2$ distance | ✔ |
| IQ-Learn (Garg et al., 2022) | Inverse SAC (Haarnoja et al., 2018b) | $r(\mathbf{s}, \mathbf{a})$ | Distribution Matching | ✘ |
| OIRL (Zolna et al., 2020) | Q-weighted BC | $r(\mathbf{s}, \mathbf{a})$ | Distribution Matching | ✔ |
| ValueDice (Kostrikov et al., 2019) | Weighted BC | - | DICE | ✔ |
| Demodice (Kim et al., 2022) | Weighted BC | - | DICE | ✔ |
| SMODICE (Ma et al., 2022a) | Weighted BC | - | DICE | ✔ |
| ABC (Sasaki and Yamashina, 2021) | Adversarial Learning | - | - | ✘ |
| Noisy BC (Sasaki and Yamashina, 2021) | Behavior Cloning | - | - | ✘ |
| CEIL (Liu et al., 2023a) | Hindsight Information Correction | $\mathbf{z}^*$ | Latent Expert Distribution Correction | ✘ |
| **ADR** (ours) | Density Weighted BC | $\hat{P}(\mathbf{a}|\mathbf{s})$ and $P^*(\mathbf{a}|\mathbf{s})$ | ADE+ELBO of VAE | ✔ |

denotes ground truth reward. We do not compare ADR with ABC and Noisy BC because our ablations (`Max ADE`, `Noisy Test`) have included settings with similar objectives.

## 7.1 MAJORITY EXPERIMENTAL RESULTS

Table 2: **Experimental results of Kitchen and Androits domains.** We test ADR on androits and kitchen domains and average the normalized D4rl score across multiple seeds. In particular, the experimental results of BC, CQL (oracle), and IQL (oracle) are directly quoted from Kostrikov et al. (2021), and results of IQL (OTR) on adroit domain are directly quoted from Luo et al., where oracle denotes ground truth reward.

| IL Tasks (`LfD (1)`) | BC | CQL (oracle) | IQL (oracle) | IQL (OTR) | IQL (CLUE) | ADR |
|---|---|---|---|---|---|---|
| `door-cloned` | 0.0 | 0.4 | 1.6 | 0.01 | 0.02 | **4.8±1.1** |
| `door-human` | 2 | 9.9 | 4.3 | 5.92 | 7.7 | **12.6±3.9** |
| `hammer-cloned` | 0.6 | 2.1 | 2.1 | 0.88 | 1.4 | **17.6±3.3** |
| `hammer-human` | 1.2 | 4.4 | 1.4 | 1.79 | 1.9 | **21.7±11.8** |
| `pen-cloned` | 37 | 39.2 | 37.3 | 46.87 | 59.4 | **84.4±19.2** |
| `pen-human` | 63.9 | 37.5 | 71.5 | 66.82 | 82.9 | **120.6±10.3** |
| `relocate-cloned` | -0.3 | -0.1 | -0.2 | -0.24 | -0.23 | **-0.2±0.0** |
| `relocate-human` | 0.1 | 0.2 | 0.1 | 0.11 | 0.2 | **2.0±1.4** |
| *Total (Android)* | 104.5 | 93.6 | 118.1 | 122.2 | 153.3 | **263.5** |
| `kitchen-mixed` | 51.5 | 51.0 | 51.0 | 50.0 | - | **87.5±1.8** |
| `kitchen-partial` | 38.0 | 49.8 | 46.3 | 50.0 | - | **80.6±2.7** |
| `kitchen-completed` | 65.0 | 43.8 | 62.5 | 50.0 | - | **95.0±0.0** |
| *Total (Kitchen)* | 104.5 | 144.6 | 159.8 | 150.0 | - | **263.1** |
| **Total (Kitchen&Android)** | 259 | 238.2 | 277.9 | 272.2 | - | **526.6** |

**LfD on Androits and kitchen domains.**  We test ADR on tasks sourced from Adroit and Kitchen domains. In particular, during the training process, we utilize single trajectory with the highest Return as a demonstration. The experimental results are summarized in Table 2, ADR achieves an impressive summed score of **526.6** points, representing an improvement of **89.5%** compared to IQL (oracle), **121.1%** compared to CQL (oracle), and surpassing all IL baselines, thus showcasing its competitive performance in long-horizon IL tasks. Meanwhile, these competitive experimental results also validate our claim that ADR, which optimizes policy in a single-step manner, can avoid the cumulative bias associated with multi-step updates using biased reward/Q functions within the RL framework. Moreover, this experiment also indicates its feasibility to utilize ADR to conduct LfD without introducing extra datasets as demonstrations.

**LfD on Gym-Mujoco domain.**  The majority of the experimental results on the tasks sourced from the Gym-Mujoco domain are displayed in Table 3. We utilized 5 expert trajectories as demonstrations and conducted ILD on all selected tasks. ADR achieves a total of **1008.7** points, surpassing most reward estimating and Q function estimating approaches. Therefore, the performance of our approach on continuous control has been validated. In particular, 1) ADR performs better than

Table 3: **Experimental results of Gym-Mujoco domain.** We utilize 5 expert trajectories as a demonstration to conduct LfD setting IL experiment, our experimental results are averaged multiple times of runs. In particular, `m` denotes `medium`, `mr` denotes `medium-replay`, `me` denotes `medium-expert`.

| LfD (5) | ORIL (TD3+BC) | SQIL (TD3+BC) | IQ-Learn | ValueDICE | DemoDICE | SMODICE | CEIL | ADR |
|---|---|---|---|---|---|---|---|---|
| hopper-me | 51.2 | 5.9 | 21.7 | 72.6 | 63.7 | 64.7 | 80.8 | **109.1±3.2** |
| halfcheetah-me | 79.6 | 11.8 | 6.2 | 1.2 | 59.5 | 63.8 | 33.9 | **74.3±2.1** |
| walker2d-me | 38.3 | 13.6 | 5.2 | 7.4 | 101.6 | 55.4 | 99.4 | **110.1±0.2** |
| Ant-me | 6.0 | -5.7 | 18.7 | 30.2 | 112.4 | 112.4 | 85.0 | **132.7±0.3** |
| hopper-m | 42.1 | 45.2 | 17.2 | 59.8 | 50.2 | 54.1 | 94.5 | 69.0±1.1 |
| halfcheetah-m | 45.1 | 14.5 | 6.4 | 2 | 41.9 | 42.6 | 45.1 | 44.0±0.1 |
| walker2d-m | 44.1 | 12.2 | 13.1 | 2.8 | 66.3 | 62.2 | 103.1 | 86.3±1.7 |
| Ant-m | 25.6 | 20.6 | 22.8 | 27.3 | 82.8 | 86.0 | 99.8 | **106.6±0.5** |
| hopper-mr | 26.7 | 27.4 | 15.4 | 80.1 | 26.5 | 34.9 | 45.1 | **74.7±1.7** |
| halfcheetah-mr | 2.7 | 15.7 | 4.8 | 0.9 | 38.7 | 38.4 | 43.3 | 39.2±0.1 |
| walker2d-mr | 22.9 | 7.2 | 10.6 | 0 | 38.8 | 40.6 | 81.1 | 67.3±4.7 |
| Ant-mr | 24.5 | 23.6 | 27.2 | 32.7 | 68.8 | 69.7 | 101.4 | 95.4±1.1 |
| **Total (Gym-Mujoco)** | 408.8 | 192 | 169.2 | 316.9 | 751.2 | 724.7 | 912.5 | **1008.7** |

ORIL, IQL-Learn demonstrating the advantage of ADR over reward estimating+RL approaches. 2) The superior performance of ADR compared to SQIL, DemoDice, SMODICE, ValueDice highlights the density weights over other regressive forms.

## 7.2 ABLATIONS

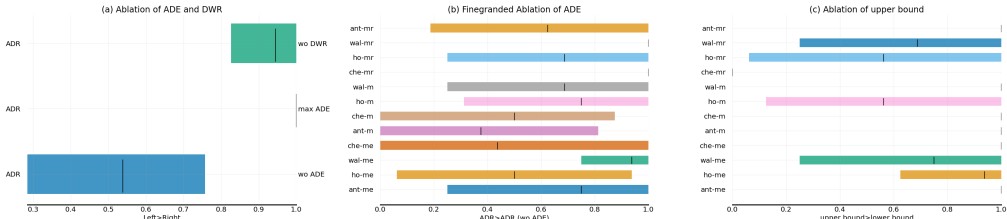

Figure 3: Ablation Results. We utilized the reliable library proposed by Agarwal et al. to conduct our experiments. The results show that the experimental setting on the left side performed better with a higher probability. Specifically, in (a) we removed part of modules *i.e.*, ADE or DWR from ADR and observed a reduction in performance. In (b), we further conducted comparisons among all tasks. Regarding (c), we carried out a fine-grained comparison of the upper and lower bounds of Equation 9 among all tasks. Note, (a) The left and right y-axes represent the selected algorithms A and B, respectively, while the x-axis represents the confidence in A>B. (b, c) involve comparisons between two algorithms, and left y-axis indicates selected tasks.

**Ablation of ADE and DWR.** To demonstrate the effectiveness of ADE, we excluded ADE *i.e.*, $J_{\text{ADE}}(\Theta^*)$ from ADR during the VAE training process. Subsequently, we optimized by maximizing the target behavior density and minimizing the sub-optimal behavior density, and we name this experimental setting as ADR (wo ADE), as shown in Figure 3 (a). ADR (wo ADE) performs better than ADR with over 50% confidence, validating the improvement brought by ADE. Meanwhile, in order to demonstrate the necessity of DWR, we 1) conducted an ablation by removing DWR, denoted as ADR (wo DWR), and found that ADR performs better than ADR (wo DWR) over 95% confidence. This indicates that DWR is necessary for ADR. 2) Optimizing the policy by solely maximizing the expression $\mathcal{L}_{\pi_\beta}(\mathbf{s}, \pi_\theta(\cdot|\mathbf{s}); \Theta^*, \Phi^*, L)|_{\mathbf{s} \sim \mathcal{D}}$, which is termed as `max ADE`, as shown in Figure 3 (a). According to the results, ADR performs better than `max ADE` with over 90% confidence. Therefore, we can't optimize the policy solely by utilizing ADE and maximizing likelihood. Besides, we observe that it won't bring an overwhelming decrease by removing ADE, therefore, we further conduct fine-grand comparison across all tasks from Gym-mujoco domain, and we observe that ADR performs better than ADR (wo ADE) across all selected `mr` tasks, but lower than 50% confidence across several `m` or `me` tasks. Therefore, ADE is essential for training with lower-quality $\hat{\mathcal{D}}$, and won't bring too much improvement for training with higher-quality $\hat{\mathcal{D}}$.

**Ablation of the upper bound of ADR.** To clearly demonstrate the necessity of Equation 11, we conducted a detailed comparison across all selected Gym-mujoco tasks. As shown in Figure 3 (c), optimizing the upper bound achieved better performance across 11 out

of 12 tasks (except for che-mr) from the Gym-mujoco domain with over 50% confidence. Therefore, it is much more effective to optimize Equation 11 rather than Equation D.1.

**Robustness to demonstrations' noisy.** In order to validate that ADR is robustness to the demonstrations' noise, we choose hop-m, wal-m, ant-m, and che-m, then adding Gaussian noisy $\Delta(\mathbf{a}) \sim \mathcal{N}(0,1)$ to demonstrations with weight $w \in \{0.1, 0.3, 0.6, 0.9\}$ *i.e.*, $\hat{\mathbf{a}} \leftarrow \mathbf{a} + w \cdot \Delta(\mathbf{a})$, and utilize the Gaussian noised action to train our policy, further observing the performance decreasing. As shown in Figure 4. ADR can be well adapt to the demonstrations' noisy. As the noise ratio increases, our method shows only a slight decline in performance on ant-m. However, there is no significant drop in performance on other tasks such as

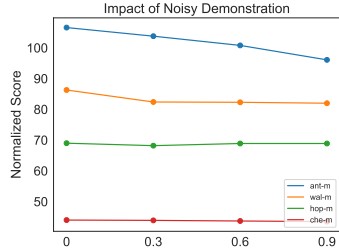

Figure 4: ADR's performance changes as the noise in the demonstrations increases.

wal-m, hop-m, and che-m. Therefore, ADR has a certain level of noise resistance and can still maintain relatively good performance even in the presence of noise within demonstrations.

**Comparison of different methods' OOD risky.** To validate our claim that ADR is a supervised in-sample IL approach and therefore does not suffer from overestimation of OOD samples, we compared three different offline algorithms, including CQL (oracle), IQL (oracle), all using the same offline datasets. Specifically, We first trained policies using four different algorithms: ADR, CQL (oracle), IQL (oracle), each with the same datasets. For example, when training ADR with $\mathcal{D}^*$ and $\hat{\mathcal{D}}$, we simultaneously trained CQL (oracle), IQL (oracle) using $\mathcal{D}^* \cup \hat{\mathcal{D}}$. After obtaining the pre-trained models, we sample states from the expert dataset and input them into these pre-trained models. We then plotted heatmaps comparing the logits obtained from these models with the expert policy showing in Figure 5. ADR maintains its decision mode as a demonstration while being less susceptible to OOD scenarios (The more similar the top-left and bottom-right corners of the heatmap are, the closer the algorithm is to the demo).

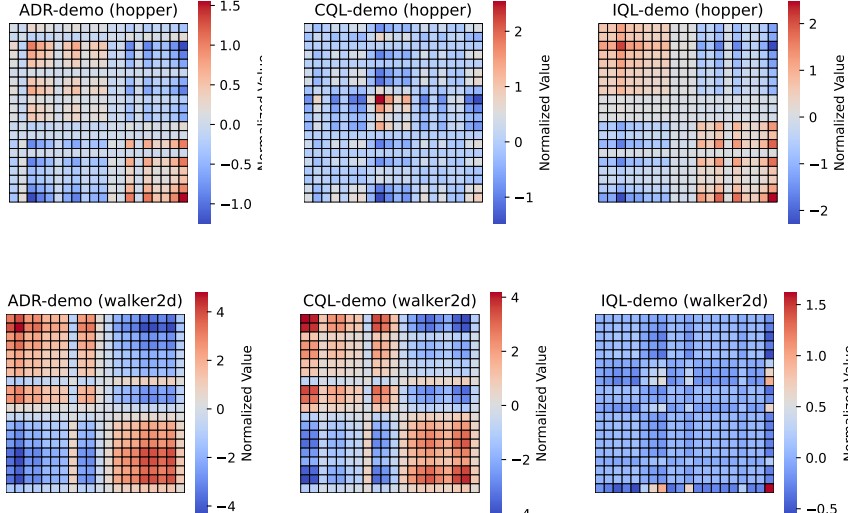

Figure 5: Heatmap of policy distributions. We stack the model's predictions alongside the samples in the dataset. The correlation is higher in the top-left and bottom-right regions, while it is lower in the other areas, the algorithm is less affected by OOD while maintain good performance (details see Appendix F).

## 8 CONCLUSION

We proposed ADR, a single-step optimization IL algorithm. Compared to traditional IL algorithms, ADR has two key advantages. First, ADR is a single-step update algorithm, and our theoretical proof shows that minimizing the ADR optimization objective is equivalent to obtaining the optimal policy, resulting in more stable training process. The second advantage is that ADR does not involve modeling the reward or value functions, so it is not affected by sub-optimal value or reward functions. To validate ADR's experimental performance, we tested it on tasks from various tasks in Android and Gym, where ADR outperformed our selected baselines.

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

CONTENTS

## A  LIMITATIONS

We have currently attempted to extend ADR to sequential models, such as the Decision Transformer (DT) (Chen et al., 2021) (Remove the Return token and use transformer as a fully supervised policy), but we have found that the experimental results are not as impressive as those under the MDP setting. We will further explore the possibility of extending ADR to sequential models.

## B  SOCIAL IMPACTS

We propose a new supervised iIL framework, ADR. Meanwhile, we point out that the advantage of ADR lies in that it can effectively avoid the cumulative offset sourced from sub-optimal Reward/Value function. Besides we In addition, the effect of ADR exceeds all previous imitation learning frameworks and even achieves better performance than IQL on robotic arm/kitchen tasks, which will greatly promote the development of imitation learning frameworks under supervised learning.

## C  HYPER PARAMETERS AND IMPLEMENTATION DETAILS

Our method is slightly dependent on hyper-parameters. We introduce the core hyperparameters here:

Table 4: Crucial hyper-parameters of ADR.

| Hyperparameter | Value |
|---|---|
| VAE training iterations | $1e^5$ |
| policy training iterations | $1e^6$ |
| batch size | 64 |
| learning rate (lr) of $\pi$ | $1e^{-4}$ |
| lr of VQ-VAE | $1e^{-3}$ |
| evaluation frequency | $1e^3$ |
| L in Equation 4 | 1 |
| $\lambda$ in Equation 15 | 1 |
| Optimizing Equation 11 | All selected tasks except for `che-mr` |
| Random Seeds | $\{0,2,4,6\}$ |
| Optimizing Equation 8 | `che-mr` |
| Model Architecture | |
| MLP Policy | $4\times$ Layers MLP (hidden dim 256) |
| VQVAE (encoder and decoder) | $3\times$ Layers MLP (hidden dim: $2\times$ action dim; latent dim: 750) |
| | 4096 tabular embeddings |

Our code is based on CORL (Tarasov et al., 2022). Specifically, in terms of a training framework, we adapted the offline training framework of Supported Policy Optimization (SPOT) (Wu et al., 2022), decomposing it into multiple modules and modifying it to implement our algorithm. Regarding the model architecture, we implemented the VQVAE ourselves, while the MLP policy architecture is based on CORL. Some general details such as warm-up, a discount of lr, *e.g.*, are implemented by CORL. *We have appended our source code in the supplement materials.*

**Computing efficiency of DWR.**  To further showcase the computational efficiency of DWR, we selected the `che-mr` environment as the benchmark and systematically varied the batch size from 10 to 300 while measuring the training time (using a 1000-step size in the policy updating stage). As depicted in Figure 6, it's evident that the training time of ADR is significantly lower compared to ADE-divergence (which shares the same conceptual framework as Equation 7), and such advantage becomes especially pronounced with larger batch sizes. Therefore, the computational efficiency of ADR has been convincingly demonstrated.

**Ablation of the upper bound of ADR.**  In order to demonstrate the effectiveness of minimizing Equation 11 (upper-bound) over minimizing Equation 8 (objective), we conduct fine-grained com-

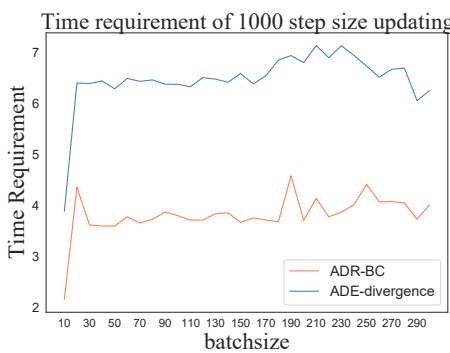 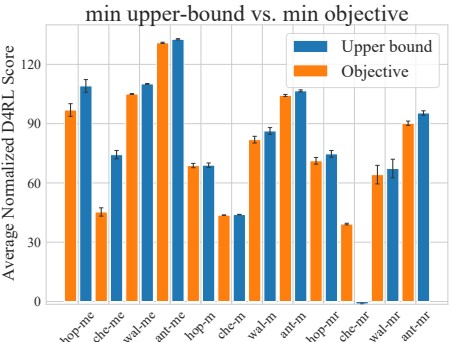

Figure 6: (Left) Comparison of training time. (Right) Abaltion of upper bound.

parisons. Specifically, we compare minimizing Equation 11, Equation 8 on all selected tasks sourced from Gym-Mujoco domain (hop denotes hopper, wal denotes walker2d, che denotes halfcheetah), minimizing Equation 11 achieve overall better performance (8 out of 12), indicating the necessity of Equation 11.

# D    THEORETICAL ANALYSIS

**Theorem D.1** (Density Weight). *Given expert log behavior density* $\log P^*(\mathbf{a}|\mathbf{s}) : \mathcal{S} \times \mathcal{A} \to \mathbb{R}$, *sub-optimal log behavior density* $\log \hat{P}(\mathbf{a}|\mathbf{s}) : \mathcal{S} \times \mathcal{A} \to \mathbb{R}$, *and the empirical policy* $\pi_\theta : \mathcal{S} \to \mathcal{A}$, *offline dataset* $\mathcal{D}$. *Minimizing the KL divergence between* $\pi_\theta$ *and* $P^*$, *while maximizing the KL divergence between* $\pi_\theta$ *and* $\hat{P}$, *i.e., Equation 7. is equivalent to:* $\min_{\pi_\theta} \mathbb{E}_{(\mathbf{s},\mathbf{a})\sim\mathcal{D}} \left[ \log \frac{\hat{P}(\mathbf{a}|\mathbf{s})}{P^*(\mathbf{a}|\mathbf{s})} \cdot ||\pi_\theta(\cdot|\mathbf{s}) - \mathbf{a}||_2 \right]$,

*Proof*

$$
\begin{aligned}
J(\pi_\theta) &= \mathbb{E}_{(\mathbf{s},\mathbf{a})\sim\mathcal{D}}[D_{\mathrm{KL}}[\pi_\theta||P^*] - D_{\mathrm{KL}}[\pi_\theta||\hat{P}]] \\
&= \mathbb{E}_{(\mathbf{s},\mathbf{a})\sim\mathcal{D}}\left[\pi_\theta(\mathbf{a}|\mathbf{s}) \cdot \log \frac{\pi_\theta(\mathbf{a}|\mathbf{s})}{P^*(\mathbf{a}|\mathbf{s})}\right] - \mathbb{E}_{(\mathbf{s},\mathbf{a})\sim\mathcal{D}}\left[\pi_\theta(\mathbf{a}|\mathbf{s}) \cdot \log \frac{\pi_\theta(\mathbf{a}|\mathbf{s})}{\hat{P}(\mathbf{a}|\mathbf{s})}\right] \\
&= \mathbb{E}_{(\mathbf{s},\mathbf{a})\sim\mathcal{D}}\left[\pi_\theta(\mathbf{a}|\mathbf{s}) \cdot \left( \log \frac{\pi_\theta(\mathbf{a}|\mathbf{s})}{P^*(\mathbf{a}|\mathbf{s})} - \log \frac{\pi_\theta(\mathbf{a}|\mathbf{s})}{\hat{P}(\mathbf{a}|\mathbf{s})} \right)\right] \\
&= \mathbb{E}_{(\mathbf{s},\mathbf{a})\sim\mathcal{D}}\left[\pi_\theta(\mathbf{a}|\mathbf{s}) \cdot \log \frac{\hat{P}(\mathbf{a}|\mathbf{s})}{P^*(\mathbf{a}|\mathbf{s})}\right] \\
&= \mathbb{E}_{(\mathbf{s},\mathbf{a})\sim\mathcal{D}}\left[\mathcal{W}(\hat{P}, P^*) \cdot \pi_\theta(\mathbf{a}|\mathbf{s})\right] \\
&\stackrel{\mathrm{def}}{=} \mathbb{E}_{(\mathbf{s},\mathbf{a})\sim\mathcal{D}}\left[\mathcal{W}(\hat{P}, P^*) \cdot ||\pi_\theta(\cdot|\mathbf{s}) - \mathbf{a}||^2\right]
\end{aligned}
\tag{17}
$$

**Assumption D.2.** *Assuming* $D_{KL}[\pi^*||\hat{\pi}] \le \delta$

**Theorem D.3.** *Given* $\mathcal{D}^*$, *based on Assumption D.2, we have:*

$$
\mathbb{E}_{\mathcal{D}^*}[\pi^* \log \frac{\pi^*}{\hat{\pi}}] \le \frac{M}{2n} \cdot \sqrt{\log \frac{2}{\delta}}
\tag{18}
$$

*with probability* $1 - \delta$. *Where* $n = |\mathcal{D}^*|$, $M = \max_{(\mathbf{s}_t, \mathbf{a}_t)} \pi^*(\mathbf{a}_t|\mathbf{s}_t) \log \frac{\pi^*(\mathbf{a}_t|\mathbf{s}_t)}{\hat{\pi}(\mathbf{a}|\mathbf{s})}|_{(\mathbf{s}_t, \mathbf{a}_t)\sim\mathcal{D}^*}$

*Proof*

Our derivation is based on Hoeffding in-equality, and We first let $X_i = \pi^*(\mathbf{a}_i|\mathbf{s}_i) \log \frac{\pi^*(\mathbf{a}_i|\mathbf{s}_i)}{\hat{\pi}(\mathbf{a}|\mathbf{s})}$, $\bar{X} = \frac{\sum_t X_t}{n}$, then we have:

$$
P(|\bar{X}_i - \mathbb{E}_{\pi^*}[D_{KL}[\pi||\pi^*]]| \ge m) \le 2 \cdot e^{-\frac{2n^2 \cdot m^2}{M^2}}
\tag{19}
$$

Then let $2 \cdot e^{-\frac{2n^2 \cdot m^2}{M^2}} = \delta$, we obtain $t = \frac{M}{2n} \sqrt{\log \frac{2}{\delta}}$. Furthermore, with $1 - \delta$ probability we have:

$$|\bar{X}_i - \mathbb{E}_{\pi^*}[D_{KL}[\pi||\pi^*]]| \le 2 \cdot e^{-\frac{2n^2 \cdot m^2}{M^2}} \tag{20}$$

Meanwhile, we have assumed that $D_{KL}[\pi^*||\hat{\pi}] \le \delta$, and thus we obtain $\mathbb{E}_{\mathcal{D}^*}[\pi^* \log \frac{\pi^*}{\hat{\pi}}] \le \frac{M}{2n} \cdot \sqrt{\log \frac{2}{\delta}}$

**Proposition D.4** (Policy Convergence of ADR). *Assuming Equation 7 can finally converge to $\epsilon$ via minimizing Eq 9, meanwhile, assuming Assumption D.2 is held. Then* $\mathbb{E}_{(\mathbf{s},\mathbf{a}) \sim \hat{\mathcal{D}}}[D_{KL}(\pi||\pi^*)] \to \frac{M}{2n} \cdot \sqrt{\log \frac{2}{\delta}} + \Delta C + \epsilon$. *Where* $n = |\mathcal{D}^*|, M := \arg\max_{X_i}\{X_i = \pi^*(\mathbf{a}_t|\mathbf{s}_t) \log \frac{\pi^*(\mathbf{a}_t|\mathbf{s}_t)}{\hat{\pi}(\mathbf{a}_t|\mathbf{s}_t)}|(\mathbf{s}_t, \mathbf{a}_t) \sim \mathcal{D}^*\}$ *with probability* $1 - \delta$.

*Proof*

Using Bayes' rule, we have: $P^*(\mathbf{a}|\mathbf{s}) = \frac{\pi^*(\mathbf{a}|\mathbf{s})P(\mathbf{s})}{P^*(\mathbf{s})}, \quad \hat{P}(\mathbf{a}|\mathbf{s}) = \frac{\hat{\pi}(\mathbf{a}|\mathbf{s})P(\mathbf{s})}{\hat{P}(\mathbf{s})}$

Substitute it into the KL divergence terms in the objective function. $D_{KL}[\pi||P^*]$, $D_{KL}[\pi||\hat{P}]$, we have

$$\mathbb{E}_{\mathcal{D}}[D_{KL}[\pi||P^*]] = \mathbb{E}_{\mathcal{D}}\left[\pi(\mathbf{a}|\mathbf{s}) \cdot \log \frac{\pi(\mathbf{a}|\mathbf{s})}{P^*(\mathbf{a}|\mathbf{s})}\right] = \mathbb{E}_{\mathcal{D}}[D_{KL}[\pi||\pi^*]] + C_1 \tag{21}$$

$$\mathbb{E}_{\mathcal{D}}[D_{KL}[\pi||\hat{P}]] = \mathbb{E}_{\mathcal{D}}\left[\pi(\mathbf{a}|\mathbf{s}) \cdot \log \frac{\pi(\mathbf{a}|\mathbf{s})}{\hat{P}(\mathbf{a}|\mathbf{s})}\right] = \mathbb{E}_{\mathcal{D}}[D_{KL}[\pi||\hat{\pi}]] + C_2 \tag{22}$$

Here, $C_1$ and $C_2$ are constants related to the marginal distribution of the state $P(s)$, $\hat{P}(s)$ and $P^*(s)$, and they do not change with the policy $\pi$

Then, we bring Equation 21 and Equation 22 to Equation 7. Then we have

$$\mathbb{E}_{\mathcal{D}}[D_{KL}[\pi||\pi^*]] + C_1 - (\mathbb{E}_{\mathcal{D}}[D_{KL}[\pi||\hat{\pi}]] + C_2) \le \epsilon \tag{23}$$

**Case 1**  Meanwhile, we can observe from Equation D.1 that it's a weighted BC objective, and we assume this objective can well estimate the offline dataset *i.e.*, $\mathbb{E}_{\hat{\mathcal{D}}}[D_{KL}[\pi||\hat{\pi}]] \to 0$, therefore $\mathbb{E}_{\mathcal{D}}[D_{KL}[\pi||\hat{\pi}]] = \mathbb{E}_{\hat{\mathcal{D}} \cup \mathcal{D}^*}[D_{KL}[\pi||\hat{\pi}]] \approx \mathbb{E}_{\hat{\mathcal{D}}}[D_{KL}[\pi||\hat{\pi}]]$.

**Case 2**  Similar to **Case 1**, we can also obtain: $\mathbb{E}_{\mathcal{D}^*}[D_{KL}[\pi||\hat{\pi}]] \approx \mathbb{E}_{\mathcal{D}^*}[D_{KL}[\pi^*||\hat{\pi}]]$.

Assign Equation 23, we have

$$\mathbb{E}_{\mathcal{D}}[D_{KL}[\pi||\pi^*]] - \mathbb{E}_{\mathcal{D}}[D_{KL}[\pi||\hat{\pi}]] \le \epsilon + C_2 - C_1 \tag{24}$$

$$\textbf{(Pinsker's in-equality) } \mathbb{E}_{\mathcal{D}}[D_{KL}[\pi||\pi^*]] \le \mathbb{E}_{\mathcal{D}}[D_{KL}[\pi||\hat{\pi}]] + \Delta C + \epsilon \tag{25}$$

$$(\textbf{Case 1}) \ \mathbb{E}_{\mathcal{D}}[D_{KL}[\pi||\pi^*]] \le \mathbb{E}_{\mathcal{D}^*}[D_{KL}[\pi||\hat{\pi}]] + \Delta C + \epsilon \tag{26}$$

$$(\textbf{Case 2}) \ \mathbb{E}_{\mathcal{D}}[D_{KL}[\pi||\pi^*]] \le \mathbb{E}_{\mathcal{D}^*}[D_{KL}[\pi^*||\hat{\pi}]] + \Delta C + \epsilon \tag{27}$$

$$(\textbf{Theorem } D.3) \ \mathbb{E}_{\mathcal{D}}[D_{KL}[\pi||\pi^*]] \le \frac{M}{2n} \cdot \sqrt{\log \frac{2}{\delta}} + \Delta C + \epsilon, \tag{28}$$

where, $\Delta C = C_1 - C_2$ is a constant term, dependent on the state distribution. $\delta$ originates from Assumption D.2, $n = |\mathcal{D}^*|$, $M := \arg\max_{X_i}\{X_i = \pi^*(\mathbf{a}_t|\mathbf{s}_t) \log \frac{\pi^*(\mathbf{a}_t|\mathbf{s}_t)}{\hat{\pi}(\mathbf{a}_t|\mathbf{s}_t)}|(\mathbf{s}_t, \mathbf{a}_t) \sim \mathcal{D}^*\}$.

**Lemma D.5.** *Given the state distribution of empirical and expert policy $d(\mathbf{s})$, $d^{\pi^*}(\mathbf{s})$. Meanwhile, given the state-action distribution of empirical and expert policy $d^{\pi}(\mathbf{s}, \mathbf{a})$, $d^{\pi^*}(\mathbf{s}, \mathbf{a})$ we have:*

$$D_{KL}[d^{\pi}(\mathbf{s})||d^{\pi^*}(\mathbf{s})] \le D_{KL}[d^{\pi}(\mathbf{s}, \mathbf{a})||d^{\pi^*}(\mathbf{s}, \mathbf{a})] \tag{29}$$

**Lemma D.6.** *Given the distribution of empirical and expert transitions $d^\pi(\mathbf{s}, \mathbf{a}, \mathbf{s}')$, $d^{\pi^*}(\mathbf{s}, \mathbf{a}, \mathbf{s}')$ we have following relationship:*

$$D_{KL}[d^\pi(\mathbf{s}, \mathbf{a}, \mathbf{s}')||d^{\pi^*}(\mathbf{s}, \mathbf{a}, \mathbf{s}')] = D_{KL}[d^\pi(\mathbf{s}, \mathbf{a})||d^{\pi^*}(\mathbf{s}, \mathbf{a})] \tag{30}$$

*Proof* of Lemma D.5 and Lemma D.6 see Lemma 1 and Lemma 2 from Ma et al.

**Assumption D.7.** *Suppose the policy extracted from Equation is $\pi$, we separately define the state marginal of the dataset, empirical policy, and expert policy as $d^{\mathcal{D}}$, $d^\pi$ and $d^{\pi^*}$, they satisfy this relationship:*

$$D_{KL}[d^\pi||d^{\pi^*}] \le D_{KL}[d^{\mathcal{D}}||d^{\pi^*}] \tag{31}$$

**Lemma D.8** (lemma 2 from Cen et al. (2024)). *Suppose the maximum reward is $R_{max} = \max ||r(\mathbf{s}, \mathbf{a})||$, and $V(\rho_0) = \mathbb{E}_{\mathbf{s}_0}[V(\mathbf{s}_0)]$ denote the performance given a policy $\pi$, then with Assumption D.7:*

$$|V^\pi(\rho_0) - V^{\pi^*}(\rho_0)| \le \frac{R_{max}}{1 - \gamma} D_{TV}[d^*(\mathbf{s})||d^{\mathcal{D}}(\mathbf{s})] + \frac{2 \cdot R_{max}}{1 - \gamma} E_{d^{\mathcal{D}}}[D_{TV}[\pi(\cdot|\mathbf{s})||\pi^*(\cdot||\mathbf{s})]] \tag{32}$$

Proof of Lemma D.8 see Lemma 2 from Cen et al.

**Proposition D.9.** *(Value Bound of ADR) Given the empirical policy $\pi$ and the optimal policy $\pi^*$, let $V^\pi(\rho_0)$ and $V^{\pi^*}(\rho_0)$ separately denote the value network of $\pi$ and $\pi^*$, and given the discount factor $\gamma$. Meanwhile, let $R_{max}$ as the upper bound of the reward function i.e., $R_{max} = \max ||r(\mathbf{s}, \mathbf{a})||$. Based on the Assumption D.7, Assumption D.2, Lemma D.8, and Proposition 5.2, we can obtain:*

$$|V^\pi(\rho_0) - V^{\pi^*}(\rho_0)| \le \frac{R_{max}}{1 - \gamma} D_{TV}[d^*(\mathbf{s})||d^{\mathcal{D}}(\mathbf{s})] + \frac{2 \cdot R_{max}}{1 - \gamma} \cdot \sqrt{2 \cdot (\frac{M}{2n} \cdot \sqrt{\log \frac{2}{\delta}} + \Delta C + \epsilon)}, \tag{33}$$

*Where, $\Delta C = C_1 - C_2$ is a constant term, typically dependent on the state distribution. The $\delta$ originates from Assumption D.2, $n = |\mathcal{D}^*|$, $M := \arg\max_{X_i}\{X_i = \pi^*(\mathbf{a}_t|\mathbf{s}_t) \log \frac{\pi^*(\mathbf{a}_t|\mathbf{s}_t)}{\hat{\pi}(\mathbf{a}_t|\mathbf{s}_t)} | (\mathbf{s}_t, \mathbf{a}_t) \sim \mathcal{D}^*\}$.*

*Proof*

In Proposition 5.2, we have proved that if $\mathbb{E}_{(\mathbf{s},\mathbf{a})\sim\mathcal{D}}\left[\pi_\theta(\mathbf{a}|\mathbf{s}) \cdot \log \frac{\hat{P}(\mathbf{a}|\mathbf{s})}{P^*(\mathbf{a}|\mathbf{s})}\right]$ can finally converge to $\epsilon$. Then $\mathbb{E}_{(\mathbf{s},\mathbf{a})\sim\hat{\mathcal{D}}}[D_{KL}(\pi||\pi^*)] \to \frac{M}{2n} \cdot \sqrt{\log \frac{2}{\delta}} + \Delta C + \epsilon$

Subsequently, based on Lemma D.8, we derivative:

$$\begin{aligned}
|V^\pi(\rho_0) - V^{\pi^*}(\rho_0)| \le& \frac{R_{max}}{1 - \gamma} D_{TV}[d^*(\mathbf{s})||d^{\mathcal{D}}(\mathbf{s})] + \frac{2 \cdot R_{max}}{1 - \gamma} E_{d^{\mathcal{D}}}[D_{TV}[\pi(\cdot|\mathbf{s})||\pi^*(\cdot||\mathbf{s})]] \tag{34}\\
\le& \frac{R_{max}}{1 - \gamma} D_{TV}[d^*(\mathbf{s})||d^{\mathcal{D}}(\mathbf{s})] + \frac{2 \cdot R_{max}}{1 - \gamma} E_{d^{\mathcal{D}}}[\sqrt{2 \cdot D_{KL}[\pi(\cdot|\mathbf{s})||\pi^*(\cdot||\mathbf{s})]}] \tag{35}\\
=& \frac{R_{max}}{1 - \gamma} D_{TV}[d^*(\mathbf{s})||d^{\mathcal{D}}(\mathbf{s})] + \frac{2 \cdot R_{max}}{1 - \gamma} \cdot \sqrt{2 \cdot (\frac{M}{2n} \cdot \sqrt{\log \frac{2}{\delta}} + \Delta C + \epsilon)} \tag{36}
\end{aligned}$$

# E   EXPERIMENTAL RESULTS OF BASELINES

Our baselines on Gym-Mujoco domain mainly includes: ORIL (Zolna et al., 2020), SQIL (Reddy et al., 2019), IQ-Learn (Garg et al., 2022), ValueDICE (Kostrikov et al., 2019), DemoDICE (Kim et al., 2022), SMODICE (Ma et al., 2022a), and CEIL (Liu et al., 2023a). The majority of experimental results of these baselines are cited from CEIL (Liu et al., 2023a).

In terms of evaluation on kitchen or androits domains. The majority baselines include OTR (Luo et al., 2023) and CLUE (Liu et al., 2023b) that utilize reward estimating via IL approaches, and policy optimization via Implicit Q Learning (IQL) (Kostrikov et al., 2021). We also encompass Conservative Q Learning (CQL) (Kumar et al., 2020b) and IQL for comparison. Specifically, these experimental results are from:

- The experiment results of OTR and CLUE are directly cited from Luo et al. and Liu et al.
- The experimental results of CQL (oracle) and IQL (oracle) are separately cited from Kumar et al. and Kostrikov et al., and the experimental results of OTR on kitchen domain is obtained by running the official codebase https://github.com/ethanluoyc/optimal_transport_reward.

# F   EVALUATION DETAILS

We run each task multiple times, recording all evaluated results and taking the highest score from each run as the outcome. We then average these highest scores. For score computation, we use the same metric as D4rl *i.e.*, $\frac{\text{output}-\text{expert}}{\text{expert}-\text{random}} \times 100$. Our experiment are running on computing clusters with $16\times4$ core cpu (Intel(R) Xeon(R) CPU E5-2637 v4 @ 3.50GHz), and $16\times$RTX2080 Ti GPUs

Table 5: Experimental results from All seeds. Includes 5 demonstrations for learning from demonstration (Lfd) on the Gym-mujoco domain, and 1 demonstration for Lfd on the Kitchen and Androits domain. Our seeds are 0, 2, 4, 6. The training data is included in the appendix, and the value of each seed is obtained by returning the maximum value.

| Tasks | Seed 1 | Seed 2 | Seed 3 | Seed 4 | Avg. |
|---|---|---|---|---|---|
| hopper-me | 108.73135306 | 112.36561301 | 104.13708473 | 111.21583144 | 109.1± 3.2 |
| halfcheetah-me | 76.91686914 | 73.34520366 | 71.3600813 | 75.65439524 | 74.3± 2.1 |
| walker2d-me | 110.01480035 | 110.15162557 | 110.41349757 | 109.86814345 | 110.1± 0.2 |
| Ant-me | 132.47422373 | 132.43903581 | 132.87375784 | 133.18474616 | 132.7± 0.3 |
| hopper-m | 67.43902685 | 68.53755386 | 69.49494087 | 70.39486176 | 69.0± 1.1 |
| halfcheetah-m | 44.26977365 | 43.96688663 | 43.96063228 | 44.002488 | 44.0± 0.1 |
| walker2d-m | 89.01287452 | 84.82661744 | 84.96199657 | 86.20352661 | 86.3± 1.7 |
| Ant-m | 107.18757783 | 105.82195401 | 106.37078241 | 106.89800012 | 106.6± 0.5 |
| hopper-mr | 76.28604245 | 75.62349403 | 75.23570126 | 71.8023475 | 74.7± 1.7 |
| halfcheetah-mr | 39.04827579 | 39.08606318 | 39.24549748 | 39.34331542 | 39.2± 0.1 |
| walker2d-mr | 69.91171614 | 60.40786853 | 72.87922707 | 65.9015982 | 67.3± 4.7 |
| Ant-mr | 95.29014082 | 97.260068 | 94.74996758 | 94.31474188 | 95.4± 1.1 |
| door-cloned | 3.3699566 | 4.83888018 | 4.5226364 | 6.33812655 | 4.8± 1.1 |
| door-human | 9.35201591 | 13.05773712 | 9.10674378 | 18.71432687 | 12.6± 3.9 |
| hammer-cloned | 12.26944958 | 19.06662599 | 18.08395955 | 21.09296431 | 17.6± 3.3 |
| hammer-human | 9.37490127 | 13.78847087 | 40.01083644 | 23.73657046 | 21.7± 11.8 |
| pen-cloned | 110.88785576 | 92.09658 | 75.64396931 | 59.05532153 | 84.4± 19.2 |
| pen-human | 118.47072952 | 136.50561455 | 107.8325132 | 119.68575723 | 120.6± 10.3 |
| relocate-cloned | -0.19486202 | -0.18540353 | -0.25482428 | -0.23930115 | -0.2± 0.0 |
| relocate-human | 0.92621742 | 3.62704217 | 3.07594114 | 0.2939339 | 2.0± 1.4 |
| kitchen-mixed | 87.5 | 90.0 | 87.5 | 85.0 | 87.5± 1.8 |
| kitchen-partial | 80.0 | 77.5 | 85.0 | 80.0 | 80.6± 2.7 |
| kitchen-completed | 95.0 | - | - | - | 95.0 |

**Training stability of ADR.** Despite behavior cloning not being theoretically monotonic, we still present the training curve of ADR. As shown in Figure 7 and Figure 8, we averaged multiple runs and plotted the training curve, demonstrating that ADR exhibits stable training performance.

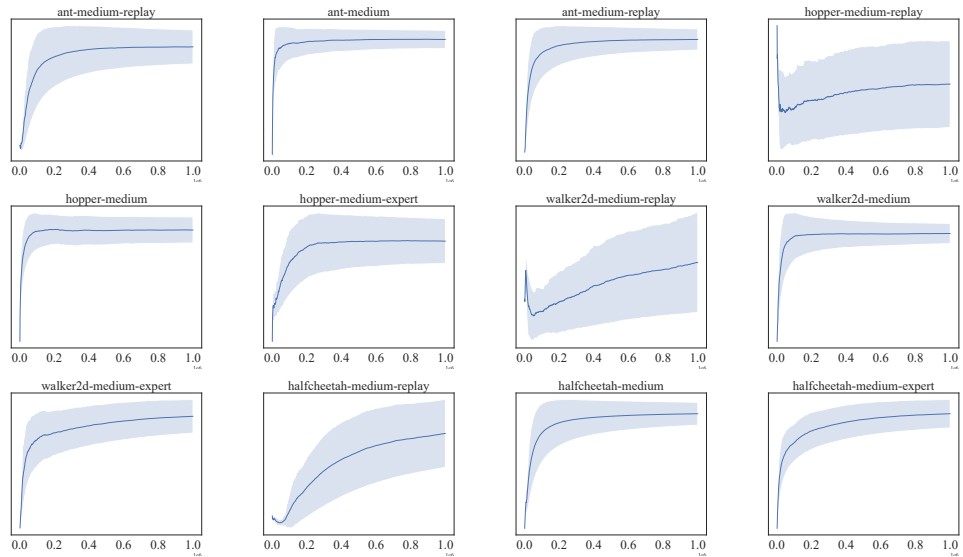

Figure 7: Training curves of ADR on all tasks sourced from Gym-Mujoco domain.

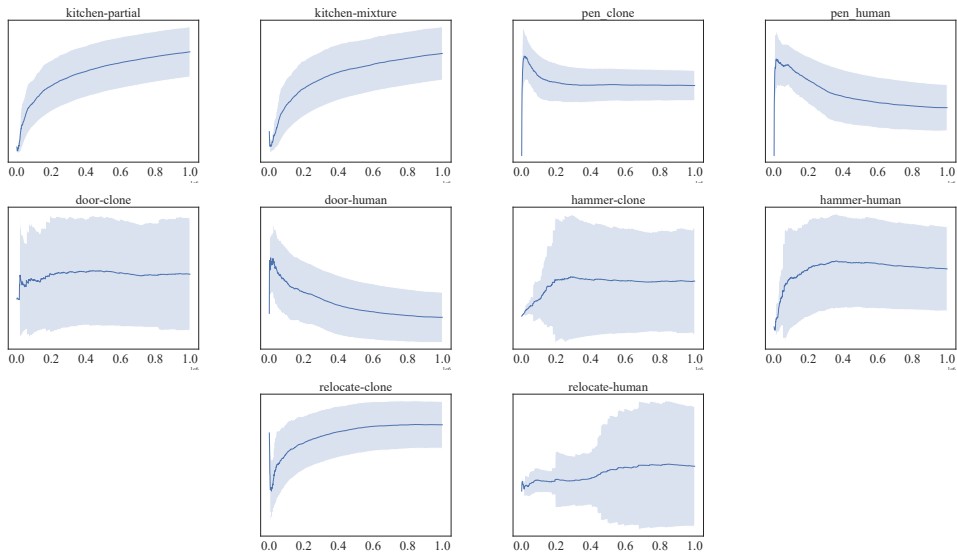

Figure 8: Training curves of ADR on tasks sourced from kitchen and androits domain.

**OOD Risky Analysis.** We further elaborate on the process of collecting experimental results related to Figure 9. Firstly, we need to train policys on chosen datasets. Specifically, our ADR is trained on five expert trajectories as demonstrations $\mathcal{D}^*$ and the complete medium-replay dataset $\hat{\mathcal{D}}$, which serves as the unknown-quality dataset mentioned in the paper, while retaining the best-performing model. Additionally, when training IQL and CQL, we mix the demonstrations $\mathcal{D}^* \cup \hat{\mathcal{D}}$ with the unknown-quality dataset and use both IQL and CQL algorithms for training. After obtaining the models, we collect the logits from different models using the following specific method: we sample the states $\{\mathbf{s}_{-20}, \mathbf{s}_{-19}, \cdots, \mathbf{s}_{-1}\} \sim \pi^*$ of the last 20 steps from a trajectory in the expert dataset and use them as inputs for ADR, IQL, and CQL. Simultaneously, we retain the actions $\{\mathbf{a}_{-20}, \mathbf{a}_{-19}, \cdots, \mathbf{a}_{-1}\} \sim \pi^*$ corresponding to these states to create heatmaps.

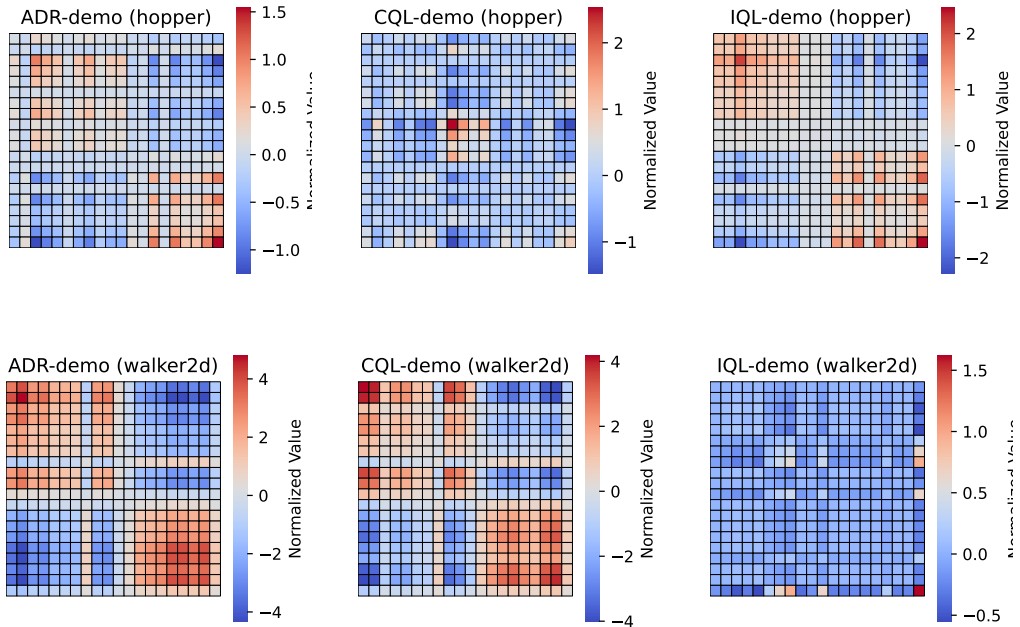

Figure 9: Heatmap of policy distributions. Higher values along the diagonal indicate a better fit of the policy to the expert policy, while lower values outside the diagonal indicate lower OOD risk for the policy.

We collect action prediction by inputting the sampled states into three models obtained by train (ADR, IQL and CQL) respectively. And after obtaining the actions, we reduce them to one dimension using PCA. Subsequently, we stack the collected actions together with the actions from the same time steps in the sampled expert dataset, calculate the covariance matrix, and then plot a heatmap to obtain Figure 9. Specifically, since the format of the dataset is `[model prediction, demo]`, only the top-left and bottom-right quarters of the heatmap have higher correlation values, which are higher than the correlations in the remaining positions of the heatmap.

For convenience, we name each heatmap plot as 'Algorithm-Demo'. From the plots, we can observe that ADR learns relatively good patterns on both the hopper and walker2d tasks, while CQL and IQL can only learn specific patterns respectively.

