# OpenReview forum: "Distribution Corrected Estimation via Adversarial Density Weighted Regression"
_ICLR.cc/2025/Conference — Submitted to ICLR 2025_

### Official Review · Reviewer_8KXJ · 2024-10-24

**Soundness:** 2
**Presentation:** 3
**Contribution:** 3
**Rating:** 6
**Confidence:** 3

**Summary:**

This paper proposes Adversarial Density Regression (ADR), a novel supervised Imitation Learning (IL) framework learning from expert and suboptimal data. Different from prior works, it minimizes the KL divergence between the learner and the expert behavior while diverging from the sub-optimal behavior. The paper proves that such objective is equivalent to weighted behavior cloning with weights being the log probability ratio on the given state-action pairs of expert and sub-optimal policy's behavior density, and estimates the weight with a VAE density estimator. The author proves the convergence of ADR theoretically and demonstrate its effectiveness on several testbeds.

**Strengths:**

1. The work is well-written and easy to follow. Though there are many mathematical notations and derivations in this paper, they are clearly introduced in a proper order. In particular, every formula in Sec. 3 and 4 is properly and briefly explained (without too much text between each formula) and highlighted with bold fonts.

2. The idea of this work is simple, intuitive but effective: it proposes a weighted behavior cloning method where the weights are probability ratio learned by VAE. The final formula of weights is straightforward, and using VAE to model distribution is a natural approach.

3. The results are not only tested on many environments (including mujoco, kitchen and adroit) with proper ablations (including loss components, upper bound of ADR, training stability and OOD risky analysis), but also guaranteed by theoretical derivations.

**Weaknesses:**

1. The literature investigation could be improved.

a) While imitation learning can indeed be categorized as mentioned in the "imitation learning" part of the related work, the paragraph did not show sufficient connection between related work and this work (e.g. adding a sentence like "our work falls into the category of offline LfD" would be much better).

b) The work claims that it is a DICE-type framework in its abstract, but there is no literature investigation for DICE at all (and as my next point stated, the summary for DICE is inaccurate). In fact, I feel that this work is very different from DICE because the foundamental idea of DICE is to match occupancies, while in this work "occupancy" is not involved; the VAE is essentially learning *policies*, not *occupancies*; they are each other's Lagrange dual variables. In fact, I suggest authors to check TAILO [5], which introduces a much simpler way (originated from DICE works such as SMODICE) for estimating *occupancy* ratio, and is also a streamlined one-step supervised framework with log probability ratio for weighted Behavior Cloning (BC) without any RL.


c) The introduction for DICE is inaccurate. Not all DICE works consider KL-divergence - SMODICE [1] uses $\chi^2$-divergence for many of its tested environments. There are also many DICE works that rely on Wasserstein distance, such as PW-DICE [2] and SoftDICE [3]. Also, the objective (Eq. 5 of their paper) of DemoDICE [4], which is one of your baseline, does not seem to fit in your Eq. 4; they have two KL terms instead of one with a linear reward term in the objective.

2. The last step of Eq. 18, which turns $\pi_\theta(a|s)$ into $\|\|\pi_\theta(\cdot|s)-a\|\|^2$, is based on the assumption of Gaussian policy which is not mentioned (consider discrete policies where actions cannot subtract). This constraint of the policy being Gaussian could be a potential limitation for the proposed method.

**Minor Issues**

1. There is no y-axis in Fig. 7 and Fig. 8, which makes the message from this figure unclear (usually judging from the figure the standard deviation is quite large, but this does not seem to be the case compared to the table in Appendix F).

2. line 98, "Latent" in ".../Latent representations" should be lower case.

3. It would be better to add "λ=0" on "ablation of ADE and DWR" part to help the readers understanding the ablation.

**References**

[1] Y. J. Ma et al. Smodice: Versatile offline imitation learning via state occupancy matching. In ICML, 2022.

[2] K. Yan et al. Offline Imitation from Observation via Primal Wasserstein State Occupancy Matching. In ICML, 2024.

[3] M. Sun et al. SoftDICE for Imitation Learning: Rethinking Off-policy Distribution Matching. ArXiv, 2021.

[4] G-H Kim et al. DemoDICE: Offline Imitation Learning with Supplementary Imperfect Demonstrations. In ICLR, 2022.

[5] K. Yan et al. A Simple Solution for Offline Imitation from Observations and Examples with Possibly Incomplete Trajectories. In NeurIPS, 2023.

**Questions:**

I have several questions:

1. Why does this work choose VAE instead of normalizing flow or diffusion model for estimation of density weight? One does not need approximation in Eq. 17 for normalizing flow, and diffusion models are stronger generative models than VAE (for ratio estimation, see diffusion DPO [1, 2]). An ablation on this would be great.

2. How does the ablation "ADR without DWR" work? From my understanding, DWR is the crucial step that retrieves the policy to learn.

3. Around line 187, the authors claim "Some off-policy offline frameworks... overly conservatism constrains the exploratory capacity of policies, limiting their ability to adapt and improve beyond the demonstrations provided." This is true. But the problem is, since your algorithm is an offline, it is necessary to overcome such issue since you will not explore beyond your dataset at all? Or are the authors arguing that they have a better inductive bias for OOD area than the common pessimisitic principle (in this case there are some more recent improvements such as MCQ [3])?

**References**

[1] B. Wallace et al. Diffusion Model Alignment Using Direct Preference Optimization. In CVPR, 2024.

[2] K. Black et al. Training Diffusion Models with Reinforcement Learning. ArXiv, 2023.

[3] J. Lyu et al. Mildly Conservative Q-Learning for Offline Reinforcement Learning. In NeurIPS, 2022.

---

> ### Author Response · Authors · 2024-11-19
> **Reply to Reviewer 8KXJ's Weaknesses section (part 1)**
>
> **wk.1.1** While imitation learning can indeed be categorized as mentioned in the "imitation learning" part of the related work, the paragraph did not show sufficient connection between related work and this work (e.g. **wk.1.2**  adding a sentence like "our work falls into the category of offline LfD" would be much better).
>
> **re (wk.1)** Thanks the reviewer for thoroughly reading the article. We mention the differences between ADR and various previous imitation learning algorithms in lines 97-99, specifically:
>
> In contrast, our ADR does not require estimating reward/valuefunctions/Latent representations and can directly utilize demonstrations to correct the policy distribution learned from a nunknown-quality dataset to mimic the expert policy.
>
> Meanwhile, one of the contributions of our paper is the proposal of a fully supervised single-step imitation learning algorithm that does not require estimating reward functions or updating based on the Bellman operator, thereby avoiding accumulated offsets in multi-step updates.
>
> **re (wk.1)** Imitation learning encompasses two major paradigms: Learning from Demonstration (LfD) and Learning from Observation (LfO). In LfD, demos are normal RL trajectory segments composed of $(s_t, a_t, r_t)$, while in LfO, demos are observation-only segments composed of $s_t$. Our actual optimization target is an objective function similar to density-weighted Behavioral Cloning (BC), where each term's input is $(s_t, a_t)$. Therefore, it cannot be directly used in the LfO setting. Intuitively, we could transform the output of the density weight into an observation-only output, but this change is more of an engineering attempt, and it cannot be directly justified academically due to the lack of a clear physical interpretation of $D_{KL}(\pi(a|s)||p(s))$.
>
> **wk.2** The work claims that it is a DICE-type framework in its abstract, but there is no literature investigation for DICE at all (and as my next point stated, the summary for DICE is inaccurate). In fact, I feel that this work is very different from DICE because the foundamental idea of DICE is to match occupancies, while in this work "occupancy" is not involved; the VAE is essentially learning policies, not occupancies; they are each other's Lagrange dual variables. In fact, I suggest authors to check TAILO [5], which introduces a much simpler way (originated from DICE works such as SMODICE) for estimating occupancy ratio, and is also a streamlined one-step supervised framework with log probability ratio for weighted Behavior Cloning (BC) without any RL.
>
>  **re (wk.2)** Thanks for your suggestions, and this question is very similar to **Reviewer jPp9's weekness 1**. However, in this paper, we have not referred to our algorithm as a derivative of DICE. Instead, we have termed our algorithm as a DICE-type (in line 20) supervised learning algorithm. Specifically,
>
>   **Why utilizing the word 'DICE type' in lines 20.** Our motivation for considering ADR as a DICE-type algorithm (in line 20) is:
> ADR also leverages a demo dataset to refine the distribution learned by the policy on other datasets, which shares a similarity in the 'regularization' with the DICE algorithm in that it incorporates a distance regularization term in the reward function to adjust the learned policy.
>
>   In addition, due to the limited word count in the abstract, we prefer to utilize concise language to summarize the main content of this paper.
>
>   **Why mentioned DICE in our paper.** Furthermore, the reason we chose DICE for comparison (lines 51~55. On the other hand...), is that DICE algorithms also use the experimental setup of "demo + other data" to train a policy that is close to the demo policy. This experimental setup is relatively close to our experimental setup. Meanwhile, some literatures [1] written from IL, and these literatures are very solid.
>
>   It is also important to note that we are aware of some DICE algorithms involve modifications to the reward function, which differ significantly from our method. Furthermore, most DICE papesr conduct theoretical analysis from the perspective of occupancy, while our paper conducts theoretical analysis from the perspective of return boundary. Therefore, we choose to use the term "DICE-type" rather than "DICE".
>
>   Given the historical context of DICE-related literature, we believe that our current writing balances the relationship beween RL/RL/IL and DICE from both the literature and experimental setup perspectives. Additionally, we have included supplementary explanations on this matter in an Appendix page of the latest version of our manuscript.

---

> ### Author Response · Authors · 2024-11-19
> **Reply to Reviewer 8KXJ's Weaknesses section (part 2)**
>
> **wk.3.1** The introduction for DICE is inaccurate. Not all DICE works consider KL-divergence - SMODICE [1] uses divergence for many of its tested environments. There are also many DICE works that rely on Wasserstein distance, such as PW-DICE [2] and SoftDICE [3]. Also, the objective (Eq. 5 of their paper) of DemoDICE [4], which is one of your baseline, does not seem to fit in your Eq. 4; they have two KL terms instead of one with a linear reward term in the objective.
>
> **re (wk.3.1)** Thanks for your suggestion. We have decided to replace D_{KL} in Equation 4 with D, where D represents a distance metric that includes both KL and W-1 distances.
>
> **wk.3.2** The last step of Eq. 18, which turns into, is based on the assumption of Gaussian policy which is not mentioned (consider discrete policies where actions cannot subtract). This constraint of the policy being Gaussian could be a potential limitation for the proposed method.**re (wk.3.2)** Given the discrete action space $\mathcal{A}$,
>
> for discrete decision-making tasks, we can replace the BC objective with a cross-entropy objective. i.e. alternating
>
> $J=E_{(s,a)\sim\mathcal{ D}}[\log\frac{\hat P}{P^*}||\pi(\cdot|s)-a||_2 ]$
>
> to $J=E_{(s,a)\sim\mathcal{ D}}[\log\frac{\hat P}{P^*}\cdot$Cross_entropy_loss$(\pi(\cdot|s), a) ]$.
>
> **wk.4** There is no y-axis in Fig. 7 and Fig. 8, which makes the message from this figure unclear (usually judging from the figure the standard deviation is quite large, but this does not seem to be the case compared to the table in Appendix F).
>
> **wk.5** line 98, "Latent" in ".../Latent representations" should be lower case.
>
> **re (wk.4 and wk.5)** Thank you for your suggestion. We have made the corresponding modifications to the manuscript.
>
> **wk.6** It would be better to add "λ=0" on "ablation of ADE and DWR" part to help the readers understanding the ablation.
>
> **re (wk.6)** Thanks for your questions. In the ablation study section, we have conducted different tests on each component of ADR. Specifically, we have labeled this setting as "wo-ADE" in Figure 3.
>
> Meanwhile, in this paper, we have adopted relatively novel aggregated RL metrics to evaluate how much confidence will decrease when the ADE term is removed from the VAE's objective function (i.e., "λ=0") compared to the original ADR objective function. Specifically, in Figure 3(a), the experimental results of the "wo-ADE" setting show a probability of over 50% but less than 55% to outperform ADR across all selected gym-mujoco tasks. Therefore, ADE has an impact on the performance of ADR. Additionally, we have compared each task in Figure 3(b) and found that the algorithm with ADE does not perform well on the medium-replay dataset, with a confidence level close to 90% or above. Meanwhile, the algorithm without ADE performs better on the medium and medium-expert datasets. Thus, ADE may provide more benefit to the algorithm on the medium/medium-expert datasets than on the medium-replay dataset.

---

> ### Author Response · Authors · 2024-11-19
> **Reply to Reviewer 8KXJ's question section (part 1)**
>
> **Q.1** Why does this work choose VAE instead of normalizing flow or diffusion model for estimation of density weight? One does not need approximation in Eq. 17 for normalizing flow, and diffusion models are stronger generative models than VAE (for ratio estimation, see diffusion DPO [1, 2]). An ablation on this would be great.
>
> **re (Q.1)** Thanks for your suggestions, but we believe that diffusion is not suitable for modeling density support in this context. Here are our reasons:
>
> Firstly, diffusion is generally used in reinforcement learning to fit trajectory segments, which is incompatible with our modeling assumptions. Our policy is an MDP policy, while the diffusion model fits a non-MDP likelihood. Consequently, there are concerns when calculating weights, as the non-MDP paradigm relies on modeling based on historical trajectories. For example, given $s_0$, the diffusion model requires input of trajectory segments $\tau_{-k:0}$ to calculate weights in a consistent manner, but such trajectories are not available to us.
>
> Therefore, the likelihood of diffusion in a non-MDP context cannot be directly multiplied with an MDP policy.
>
> To address the limitations of the above issue, we could make some insightful improvements. For example:
>
> - We could approximate the policy training as a local on-policy process. However, this would introduce new concerns. For example, given $s_0$, the diffusion model requires input of trajectory segments $\tau_{-k:0}$ to calculate weights in a consistent manner, but such trajectories are not available to us
>
> - Another approach would be to add noise and then denoise on the (s, a) samples. We have mainly tested this scheme and compared its effectiveness with our method on gym-mujoco tasks. We cannot guarantee that this approach will be effective, but we will try to conduct relevant tests as soon as possible.
>
> **Q.2** How does the ablation "ADR without DWR" work? From my understanding, DWR is the crucial step that retrieves the policy to learn.
>
> **re (Q.2)** Not using DWR (Divergence-Weighted Regularization) means that the policy is not updated through a linear objective, but instead, it is optimized directly from the starting point of the ADR (Adversarial Divergence Regularization) objective function, i.e., by minimizing $min E[D_{KL}(\pi||\pi^*)-D_{KL}(\pi||\hat\pi)$.
>
> In this paper, "ADR without DWR" is merely one option in our ablation study. Specifically, in Figure 3 of this paper, the corresponding tag is "max ADE". It can be observed that ADR has a confidence level close to 100% of outperforming "max ADE". Therefore, DWR is an important factor contributing to the effectiveness of ADR.

---

> > ### Author Response · Authors · 2024-11-19
> > **Reply to Reviewer 8KXJ's question section (part 2)**
> >
> > **Q.3.1** Around line 187, the authors claim "Some off-policy offline frameworks... overly conservatism constrains the exploratory capacity of policies, limiting their ability to adapt and improve beyond the demonstrations provided." This is true. But the problem is, since your algorithm is an offline, it is necessary to overcome such issue since you will not explore beyond your dataset at all? **Q.3.2**  Or are the authors arguing that they have a better inductive bias for OOD area than the common pessimisitic principle (in this case there are some more recent improvements such as MCQ [3])?
> >
> > **re (Q.3.1)** We believe your consideration makes sense when the demos are of high quality and considerable quantity. However, excessive noise in the demos can affect the stability of the strategy, and a limited number of demos can impact the robustness of the strategy. Therefore, we do not think conservatism suits all conditions. As shown in Figure 4, our method demonstrates strong resistance to noise. Meanwhile, in Table 2, our method outperforms various imitation learning algorithms when there is only one demo and also surpasses IQL using GT reward. This indicates that our method has a relatively low dependence on the number of demos, which supports our viewpoint.
> >
> > **re (Q.3.2)** Thanks for your suggestions, which we believe are extremely valuable. However, the OOD (out-of-distribution) issue the reviewer refers to is not the typical OOD problem encountered in reinforcement learning (RL). Instead, the OOD the reviewer mentions should refer to samples that are not included in the dataset but are crucial for RL performance.
> >
> > The OOD problem in RL generally arises from the extrapolation error introduced when calculating $a'=\pi(s')$ for Bellman backups during Q-learning updates. However, unlike off-policy algorithms, ADR has a lower probability of encountering OOD samples during training because it does not require calculating $a'=\pi(s')$ like Q-learning. Certainly, some IL algorithms based on IQL do not have OOD state-action pairs. Our paper focuses more on this type of OOD problem. Meanwhile, the heatmap presented in Figure 5 demonstrates a high correlation between the policy learned by ADR and the expert policy, which supports this claim.
> >
> > We believe it also makes sense to analyze the OOD issue mentioned by the reviewer from the perspective of inductive bias. Currently, our approach is a fully supervised model, so during the learning of the s->a mapping, predictions for so-called OOD samples are more likely to be based on the knowledge induced from the dataset.

---

> ### Comment · Reviewer_8KXJ · 2024-11-21
>
> Thanks for your detailed response. Here are my follow-up questions:
>
> **1. Regarding DICE and DICE-type**
>
> There are still some problems with the definition of DICE and DICE-type.
>
> The authors mention in their response that the reason for choosing DICE for comparison is that DICE use “demo+other data” to train a policy that is close to the demo policy. However, such types of problem should be categorized as “imitation learning with auxiliary imperfect demonstrations”, which is a far wider area than DICE methods. To name a few counter examples:
>
> **Methods that are not DICE but use auxiliary, imperfect demonstrations:** DWBC [1], ORIL [2], ISW-BC [3], TAILO [4];
>
> **Methods that are DICE and offline imitation learning, but did not use auxiliary, imperfect demonstrations:** SoftDICE [5].
>
> Therefore, I would say it is inappropriate to call “imitation learning with auxiliary imperfect demonstrations” to be “DICE-type”.
>
> Furthermore, if the feature for DICE-type is that it “incorporates a distance regularization term in the reward function to adjust the learned policy”, we can also say that today’s RLHF is DICE-type (see Eq. 3 in the DPO paper), which I think misses the point. Following the same logic, we can also say that TD3+BC is DICE-type, which is obviously not acknowledged by most of the RL community. Thus, the word for corporates a distance regularization term is, in my opinion, better be called “regularization” than “correction”.
>
> In fact, we can check the notion of “DICE” in the paper it is proposed [6]. In such paper, DICE is proposed for accounting for off-policy bias, essentially an **importance sampling on occupancies**, and at least as far as I know, all DICE papers are focused on state/state-pair/state-action occupancies which are not optimized in this paper.
>
> To sum up, different from reviewer jPp9, I am not so concerned on whether DICE is reinforcement learning or imitation learning – after all, as I said in my review, value functions and state occupancies are Lagrange dual variables for policy optimization (see page 11 in https://nanjiang.cs.illinois.edu/files/cs542f22/note1.pdf), and IL can be converted to RL quite smoothly. But I am concerned that the paper did not give accurate intuitions for DICE methods and did not accurately position itself in the related works.
>
> **2. Regarding ADR’s robustness to OOD problem.**
>
> The author mentioned that “ADR has a lower probability of encountering OOD samples”
> , and “The OOD problem in RL generally arises from the extrapolation error introduced when calculating $a’=\pi(s’)$ for Bellman backups during Q-learning updates.” My problem is: this is indeed a problem, but isn’t ADR also haunted with OOD problem when you need to learn policy’s log probability when you are modeling expert policy? I understand that ADE is proposed for alleviating this issue, but you also mentioned that even without ADE loss, the performance can also be good. Can the authors explain this?
>
> **References**
> [1] H. Xu et al. Discriminator-Weighted Offline Imitation Learning from Suboptimal Demonstrations. In ICML, 2022.
>
> [2] K. Zolna et al. Offline Learning from Demonstrations and Unlabeled Experience. In Offline RL Workshop @ NeurIPS, 2020.
>
> [3] Z. Li et al. Imitation Learning from Imperfection: Theoretical Justifications and Algorithms. In NeurIPS, 2023.
>
> [4] K. Yan et al. A Simple Solution for Offline Imitation from Observations and Examples with Possibly Incomplete Trajectories. In NeurIPS, 2023.
>
> [5] M. Sun et al. SoftDICE for Imitation Learning: Rethinking Off-policy Distribution Matching. ArXiv, 2021.
>
>
> [6] O. Nachum et al. DualDICE: Behavior-Agnostic Estimation of Discounted Stationary Distribution Corrections. In NeurIPS, 2019.

---

> ### Author Response · Authors · 2024-11-22
> **Reply to Reviewer 8KXJ's followed questions**
>
> **Q.1** Therefore, I would say it is inappropriate to call “imitation learning with auxiliary imperfect demonstrations” to be “DICE-type”.
>
> **re (Q.1)** Thanks very much for your suggestion. After reading your response, we agree that modifying the term "DICE-type" is reasonable. We have made the necessary changes to this term and will update our manuscript accordingly.
>
> **Q.2** Thus, the word for corporates a distance regularization term is, in my opinion, better be called “regularization” than “correction”.
>
> **re (Q.2)** Thanks for your analysis. We have revised the term used in the rebuttal process from "correction" to "regularization".
>
> **Q.3** But I am concerned that the paper did not give accurate intuitions for DICE methods and did not accurately position itself in the related works.
>
> **re (Q.3)** Thanks for the your follow-up question. Indeed, in the introduction of the related work, this paper only elaborates on the progress of past imitation learning without explicitly referencing the motivation of this paper or comparing ADR with past imitation learning algorithms.
>
> The advantage of ADR compared to previous IL algorithms is that ADR updates the policy through a single-step weighted BC form. ADR does not require shaping rewards or Q functions, so it will not suffer from the accumulated offsets stemming from suboptimal rewards that previous IL algorithms may encounter. Additionally, ADR does not need to estimate Q or rewards, so it does not require alternating updates of rewards, Q, and policies, as previous IL algorithms did.
>
> Regarding DICE, we believe it aligns closely with our experimental setup, and past research has demonstrated its high sample efficiency. Specifically, in this paper (lines 51~52), we mention "another approach termed distribution corrected estimation (DICE)...". Therefore, from our perspective, DICE can be used as a baseline for experimental performance to compare against our proposed solution. Meanwhile, as the reviewer has pointed out, from a theoretical standpoint, our solution and DICE are two distinct approaches. We will provide the necessary clarification in the supplementary pages or the related work section. Thank you.
>
> **We will make the necessary revisions in the manuscript and will indicate the locations of the changes in our public official comments once the revisions are complete.**
>
> **Q.4.1** My problem is: this is indeed a problem, but isn’t ADR also haunted with OOD problem when you need to learn policy’s log probability when you are modeling expert policy? **Q.4.2** I understand that ADE is proposed for alleviating this issue, but you also mentioned that even without ADE loss, the performance can also be good. Can the authors explain this?
>
> **re (Q.4.1)** Thanks for the your question. Simply estimating the expert's density is not sufficient to fully overcome the out-of-distribution (OOD) issue due to our demos. However, ADR does not estimate samples outside the dataset, so this part of the OOD issue does not exist. Meanwhile, in this paper, we did not refer to $P^∗$ as the expert density but instead as the good density (lines 202) to avoid this ambiguity. If we had claimed $P^*$ as the expert density, our approach would be OOD.
>
> **re (Q.4.2)** Thanks for your question. We have provided an analysis in **wk.6** showing that ADE can be more beneficial to the medium-replay dataset, which has poorer data quality. However, its benefits to the medium and medium-expert datasets are limited because their performance is already sufficiently good and close to the expert level, resulting in relatively smaller improvements. For the medium-replay dataset, the data is more scarce and diverse. The presence of ADE can significantly distinguish the datasets in medium-replay that are close to the expert from those that diverge, allowing for better capture of the expert-like segments and thus benefiting the medium-replay dataset.

---

> ### Comment · Reviewer_8KXJ · 2024-11-23
>
> Thank you for your detailed response. I appreciate the authors' effort in improving the paper. Nevertheless, given our previous discussion, I still feel the story of this paper is a bit too biased towards DICE. More specifically:
>
> 1) The current title still imply the proposed method is a DICE method;
>
> 2) since the work is not based on Eq. (4) or occupancy derivation (currently, there is no word "DICE" appearing in Sec. 4-6), presenting (4) seems unnecessary in the preliminary section, despite that it is a line of related work.

---

> ### Author Response · Authors · 2024-11-24
> **Reply to Reviewer 8KXJ's followed suggestions**
>
> Thank you very much for continuing to follow up on these suggestions. We agree with your point of view, and our paper has been revised accordingly.
>
> - We have changed the paper title to ‘Imitating from auxiliary imperfect demonstrations via Adversarial Density Weighted Regression’.
>
> - We have removed the content related to DICE in the preliminary section, and we have moved Definition 1 to the problem formulation.

---

> > ### Comment · Reviewer_8KXJ · 2024-11-24
> >
> > Thanks for your timely reply. I shall now increase my score to 6, as I feel the methodology and evaluation of the work is sufficient for ICLR. I also reduced my confidence score to 3 as I would leave the decision to AC whether the current amount of modification for the story requires another round of review or not.

---

> > > ### Author Response · Authors · 2024-11-24
> > > **Thank you very much!**
> > >
> > > Thank you very much for taking the valuable time to provide your precious suggestions. We believe that our manuscript has benefited greatly from your valuable suggestions and has become more comprehensive!

---

### Official Review · Reviewer_eWnu · 2024-10-28

**Soundness:** 2
**Presentation:** 2
**Contribution:** 1
**Rating:** 3
**Confidence:** 4

**Summary:**

The paper presents an offline imitation learning (IL) algorithm, where the agent lacks access to environment rewards and cannot collect new samples. Instead, it relies on two datasets: (1) expert demonstrations and (2) sub-optimal demonstrations. The scarcity of expert data, a realistic scenario in IL, makes learning expert-level policies particularly challenging in this offline setting.

The authors design their approach with two objectives in mind:
 It avoids the caveats of offline learning with the Bellman operator like TD learning or Value Distribution Corrected Estimation (ValueDICE) learning which are known to suffer from bootstrapping errors, especially in the offline case and with small datasets where OOD evaluations are inevitable.
It utilizes both the expert and the sub-optimal demonstrations with the goal of learning a policy that is as close as possible to the expert policy.

To achieve these goals, they design a policy loss function (termed ADR) that starts with expert behavioral cloning KL-div loss but adds negative KL-div loss (maximizing the KL-div) against the sub-optimal policies.

In order to learn this objective the authors suggest to learn an intermediate model of the expert policy P* and the sub-optimal empirical policy P^ with Variational Autoencoders (VAE) and use the log-ratio P*/P^ as a (log) Importance sampling weight for the policy loss. This results in a simple two-steps optimization problem (1) learn VAEs, (2) learn policy and avoids the Bellman operator.

In order to compensate for the scarcity of expert demonstrations, they add to the VAE loss a discriminator loss (GAIL [1] like, termed ADE) that lets adding samples from the sub-optimal dataset at the expense of altering the expert policy estimation.

The paper then analyses some theoretical bounds on the expected learned policy and its overall performance (i.e. value function).

In terms of experiments, first, the authors demonstrate some similarity of the frequency of states in the learned policy with respect to the expert policy. Then they conduct experiments in 3 test benches (Mujoco, Androit, Kitchen) and show superiority of their method over algorithms like Behavioral Cloning (BC), CQL, IQL+Oracle, IQL+OTR, IQL+CLUE which are designed for similar (yet not always identical) settings (offline RL, potentially with reward and without sub-optimal datasets) .

References:
1. Ho, Jonathan, and Stefano Ermon. "Generative adversarial imitation learning." Advances in neural information processing systems 29 (2016).

**Strengths:**

The paper clearly motivates the problem, reviews relevant methods, and identifies their limitations.

The proposed approach is straightforward and avoids the complexities of Bellman-based TD learning.

**Weaknesses:**

1. Is ADR an appropriate optimization objective?

It is questionable whether the ADR loss is an adequate objective function for this RL setting. First, intuitively when the sub-optimal policy gets closer (or converges) to the expert policy, ADR converges to a degenerated objective function. Next, even if the sub-optimal policies are sufficiently different from the expert policy, the ADR policy does not converge to the P* solution or even sufficiently close to the P* solution. For example in the discrete case, the ADR objective is:

pi = argmin sum_i pi_i \log(\frac{P^_i}{P*_i})

Lets take for example the case where P* = (0.6, 0.2, 0.2) and P^=(0.2, 0.2, 0.6) (no states in this case, only 3 actions). The ADR policy converges to pi = (1, 0, 0), since actions with similar probability  between P* and P^ do not add weight to the policy. This means that ADR tends to increase the actions with positive ratio (where P*>P^), reduce (up to zero) the actions with negative ratio (P^ > P*) and ignore the actions where P*~=P^ which fails to capture the expert policy.

Moreover, ADR tends to yield policies with smaller action-support than the expert policy (more deterministic, where some expert actions are never taken) this can lead to potentially sub-optimal results as some actions may be crucial for functional policy. This trend of smaller support of ADR can be both observed in my toy example and in Fig 2 where we find that in both ANT and HALFCHEETAH ADR has smaller state support than the expert policy.

Proposition 5.2 tries to upper-bound the KL distance between the learned policy and the expert policy. As indicated in this analysis, when the distance between P* and P^ is small (delta) then the bound gets looser.

2. Are P* and P^ really needed and does ADR really avoid bootstrapping and OOD errors?

Given that one wish to design the ADR policy, it is not clear why the learned P* and P^ models are required. In a sense, if data is sufficient we can sample the backward KL-div D-KL(P*,pi) and D-KL(P^,pi) from the data. Therefore, assuming that the reason for incorporating P* and P^ models is to compensate the data scarcity (i.e. using both the sub-optimal and the expert dataset to regress for the ADR policy), one must ask himself whether ADR really avoids the caveats of bootstrapping (in the sense of using an estimated quantity as part of the ground truth for another model) as the ground truth for ADR includes the estimated log-ratio log(P^/P*) which also means that we evaluate P* out-of-distribution over the sub-optimal dataset, which essentially requires another bandage (the ADE auxiliary loss).

3. Value bounds: Do they provide insight?

Proposition 5.3 tries to lower-bound the value differences between the ADR policy and the expert policy. In general for any two policies p and q we have |V_p(s0) - V_q(s0)| <= 2R_max / (1 - \gamma). For ADR, the upper bound contains several more multiplicative factors, however, since elements like \Delata C and D_TV(d*|d^D) are constant terms (i.e. not diminishing to zero) at any case, this bound is too loose to provide concrete reasoning about the similarity between pi and P*.

4. Is ADE a proper way to handle the demonstrations scarcity?

ADE is presented as a practical auxiliary loss to ADR that should mitigate the problem of small expert dataset at the expense of altering the P* network which does not represent anymore the estimation of the expert policy. I’m not sure this is a sound solution for the need to compensate for small dataset as it basically lead to a deadly tradeoff where as we increase the weight of the auxiliary loss we better avoid OOD samples but on the same time we move away from our desired function (i.e. the expert policy). There are other alternatives that should be considered here, for example to estimate the KL-div between the expert and the sub-optimal empirical policy (as it is done in [1]), this lets you train both networks from samples from both D* and D^ without altering the structure/behavior of both networks.

references:
1. Freund, Gideon Joseph, Elad Sarafian, and Sarit Kraus. "A coupled flow approach to imitation learning." International Conference on Machine Learning. PMLR, 2023.

**Questions:**

1. Regarding VAE for density estimation:
There appear to be issues in the formulation of the ELBO loss (Eq. 5), which should align with Eq. 4 in [1] (CLUE). Additionally, what motivates the choice of this conditional VAE structure (with actions as input/output and state as context) over simpler density estimators like normalizing flows? You do not leverage latent space information in the algorithm (as CLUE does), and evaluating \log⁡ P∗(a∣s) and \log P^*(a|s) requires Monte Carlo sampling.
Could you clarify this choice and how you approximate \log ⁡P∗(a∣s) and \log P^*(a|s)?

2. Regarding theorem 4.2 (theorem D.1 in the appendix). Can you clarify the last move in Eq 18 in the appendix, which probabilistic model do you assume the policy follows?

references:
1. Liu, Jinxin, et al. "Clue: Calibrated latent guidance for offline reinforcement learning." Conference on Robot Learning. PMLR, 2023.

---

> ### Author Response · Authors · 2024-11-19
> **Reply to Reviewer eWnu's Weaknesses section (part 1)**
>
> **wk.1** Is ADR an appropriate optimization objective?
> It is questionable whether the ADR loss is an adequate objective function for this RL setting. **wk.1.1** First, intuitively when the sub-optimal policy gets closer (or converges) to the expert policy, ADR converges to a degenerated objective function.**wk.1.2**  Next, even if the sub-optimal policies are sufficiently different from the expert policy, the ADR policy does not converge to the P* solution or even sufficiently close to the P* solution. For example in the discrete case, the ADR objective is:
>
> pi = argmin sum_i pi_i \log(\frac{P^_i}{P*_i})
>
> Lets take for example the case where P* = (0.6, 0.2, 0.2) and P^=(0.2, 0.2, 0.6) (no states in this case, only 3 actions). The ADR policy converges to pi = (1, 0, 0), since actions with similar probability between P* and P^ do not add weight to the policy. This means that ADR tends to increase the actions with positive ratio (where P*>P^), reduce (up to zero) the actions with negative ratio (P^ > P*) and ignore the actions where P*~=P^ which fails to capture the expert policy.
>
> **re (wk.1.1)** Thanks for your comprehensive analysis of this issue. Although your analytical approach provides us with intuitive insights, the actual conclusion is contrary to your analysis. The reasons are as follows: When the expert policy and the suboptimal policy are relatively close, the ratio $\frac{P^*}{\hat P}$ will approach 1. This implies that we are training using behavior cloning on a large dataset of expert demonstrations. To our knowledge, behavior cloning performs well on expert datasets, thus leading to a highly effective policy.
>
> **re (wk.1.2)** Your starting point is good, but you have overlooked a fact: when we use a density estimator for calculation, we are computing the probability of a sample, namely $\pi(a|s)$, where both $a$ and $s$ are given samples. Specifically, if the action we sample from the dataset is (1,0,0), this means that the probability output by $P^*$ is 0.6 and the probability output by $\hat P$ is 0.2. This implies that the policy will converge towards the expert policy.
>
> **wk.2.1** Moreover, ADR tends to yield policies with smaller action-support than the expert policy (more deterministic, where some expert actions are never taken) this can lead to potentially sub-optimal results as some actions may be crucial for functional policy. **wk.2.2** This trend of smaller support of ADR can be both observed in my toy example and in Fig 2 where we find that in both ANT and HALFCHEETAH ADR has smaller state support than the expert policy.
>
> **re (wk.2.1)** We believe that the reviewer holds high standards for the comprehensiveness of the paper. However, in reality, most theoretical analyses in imitation learning focus on whether the state occupancy of the algorithm can approximate the expert policy, rather than surpassing suboptimal policies. Furthermore, in the imitation learning (IL) setting, demo data is inherently available, so it is more important to approximate the performance of the demos. Additionally, we trust that the reviewer has some experience in offline reinforcement learning. The so-called policy improvement often stems from the concatenation of suboptimal trajectories by the Bellman operator, which can lead to cumulative offsets due to the presence of suboptimal rewards in the IL setting. However, our method does not have this concern.
>
> **re (wk.2.2)** Although our method shows limited improvement on the ANT and HALFCHEETAH tasks, we have achieved comprehensive enhancements compared to previous algorithms on long-distance tasks from various domains such as kitchen and Androit as well as other gym-mujoco tasks However, we did not mention the so-called policy improvement in the paper because we believe there are certain differences between the goals of imitation learning (IL) and reinforcement learning (RL). We hope the reviewer will fully consider this point.
>
> **wk.3** Proposition 5.2 tries to upper-bound the KL distance between the learned policy and the expert policy. As indicated in this analysis, when the distance between P* and P^ is small (delta) then the bound gets looser.
>
> **re (wk.3)** Your analysis is reasonable, which explains why ADR exhibits varying performance across different datasets. However, we believethat when the dataset quality exceeds medium-replay, the distance between suboptimal policies and demos falls within a certain range. However, this assumption is more influenced by random dataset. But, in practical situations, it is impossible for us to test with a completely random dataset. Compared to random data, we are more likely to use datasets of poorer quality as sub-optimal datasets..

---

> > ### Comment · Reviewer_eWnu · 2024-11-20
> > **regarding wk.1.1**
> >
> > If \frac{P^*}{\hat P} approaches 1, then \log \frac{P^*}{\hat P} approaches zero and therefore, the objective in equation (9) vanishes.

---

> > > ### Author Response · Authors · 2024-11-30
> > > **Dear Reviewer eWnu, we would like to further confirm whether we have addressed your concern, thank you.**
> > >
> > > Dear Reviewer eWnu,
> > >
> > > As the discussion deadline is approaching, we would like to further confirm whether we have addressed your concern. Thank you very much. Additionally, we humbly request that you consider increasing our score if we have successfully addressed your concerns. Thank you.
> > >
> > > Best Regard,
> > >
> > > Conference Authors

---

> > > > ### Comment · Reviewer_eWnu · 2024-12-01
> > > > **Thank you for your kind replay**
> > > >
> > > > I would like to thank the authors for their kind reply and effort to address the concerns of all reviewers.
> > > > It is not an easy task to manage so many questions and nuances while being attentive and respectful.
> > > >
> > > > To me, I believe that the fundamental issue in this paper is the assumption that ADR is an appropriate loss function to mix between expert and sub-expert trajectories, as I tried to convey in my review, this assumption leads to peculiar policies in many cases.
> > > >
> > > > I don't believe that in the current state of RL, one can support the validity and usefulness of an algorithm only via credible experimental results since they can be influenced by so many latent variables that can be potentially ignored. One must also make the case in favor of the algorithm through intuitive explanations, reasoning, and theoretical justifications which are missing from ADR.
> > > >
> > > > I believe that the setting and RL problem, tackled by the authors in this paper are relevant and timely for the real-world challenges that RL practitioners face and the authors should do a better job in coming up with a sound algorithm that learns from the sub-optimal trajectories while converging (with enough data) to the expert trajectory.
> > > >
> > > > therefore, I won't raise the current score but I wish the authors all the best, and l truly believe that this work can be fixed and presented in future venues.

---

> ### Author Response · Authors · 2024-11-19
> **Reply to Reviewer eWnu's Weaknesses section (part 2)**
>
> **wk.4** Are P* and P^ really needed and does ADR really avoid bootstrapping and OOD errors?
> Given that one wish to design the ADR policy, it is not clear why the learned P* and P^ models are required. In a sense, if data is sufficient we can sample the backward KL-div D-KL(P*,pi) and D-KL(P^,pi) from the data. Therefore, assuming that the reason for incorporating P* and P^ models is to compensate the data scarcity (i.e. using both the sub-optimal and the expert dataset to regress for the ADR policy), one must ask himself whether ADR really avoids the caveats of bootstrapping (in the sense of using an estimated quantity as part of the ground truth for another model) as the ground truth for ADR includes the estimated log-ratio log(P^/P*) which also means that we evaluate P* out-of-distribution over the sub-optimal dataset, which essentially requires another bandage (the ADE auxiliary loss).
>
> **re (wk.4)** Despite the reviewer's thorough analysis, the fact remains that ADR only uses the dataset for learning. ADR does not need to compute $a' = \pi(s')$ like some off-policy frameworks based on Q-learning, so it does not access out-of-distribution (OOD) state-action pairs during training. Therefore, there is no OOD issue.
>
> **wk.5.2** Value bounds: Do they provide insight?
> **wk.5.1** Proposition 5.3 tries to lower-bound the value differences between the ADR policy and the expert policy. In general for any two policies p and q we have |V_p(s0) - V_q(s0)| <= 2R_max / (1 - \gamma). For ADR, the upper bound contains several more multiplicative factors, however, since elements like \Delata C and D_TV(d*|d^D) are constant terms (i.e. not diminishing to zero) at any case, this bound is too loose to provide concrete reasoning about the similarity between pi and P*.
>
> **re (wk.5.1)** We believe that the conclusion of Proposition 5.3 is meaningful. The reviewer's concern may stem from the fact that this conclusion is not independent of the quality of the dataset. However, it is easy to consider the fact that **"can you design an offline algorithm whose performance is independent of the dataset? In other words, can it be trained to achieve the best possible effect regardless of the dataset?"** If this is not feasible, then Proposition 5.3 is reasonable.
>
> **re (wk.5.2)** Moreover, Proposition 5.3 clearly reflects a pattern: when the KL divergence between the expert policy and the suboptimal policy is lower, the bound of the inequality becomes smaller, which implies that $V^{\pi}$ is closer to $V^{\pi^*}$.
>
> **wk.6** Is ADE a proper way to handle the demonstrations scarcity?
> ADE is presented as a practical auxiliary loss to ADR that should mitigate the problem of small expert dataset at the expense of altering the P* network which does not represent anymore the estimation of the expert policy. I’m not sure this is a sound solution for the need to compensate for small dataset as it basically lead to a deadly tradeoff where as we increase the weight of the auxiliary loss we better avoid OOD samples but on the same time we move away from our desired function (i.e. the expert policy). There are other alternatives that should be considered here, for example to estimate the KL-div between the expert and the sub-optimal empirical policy (as it is done in [1]), this lets you train both networks from samples from both D* and D^ without altering the structure/behavior of both networks.
>
> **re (wk.6)** Thanks for your question. In fact, we have already conducted ablation experiments in this paper, with the analysis located on lines 479-482. As shown in Figure 3(b), when we remove ADE, ADR performs with over 60% confidence better than ADR (without ADE) on medium-replay, while there is a relatively minor improvement on medium-expert. This demonstrates that ADE plays a more significant role on suboptimal datasets.

---

> ### Author Response · Authors · 2024-11-20
> **Reply to Reviewer eWnu's question section**
>
> **Q.1.1** Regarding VAE for density estimation: There appear to be issues in the formulation of the ELBO loss (Eq. 5), which should align with Eq. 4 in [1] (CLUE). **Q.1.2**  Additionally, what motivates the choice of this conditional VAE structure (with actions as input/output and state as context) over simpler density estimators like normalizing flows? You do not leverage latent space information in the algorithm (as CLUE does), and evaluating \log⁡ P∗(a∣s) and \log P^*(a|s) requires Monte Carlo sampling. Could you clarify this choice and how you approximate \log ⁡P∗(a∣s) and \log P^*(a|s)?
>
> **re (Q.1.1)** Thanks for your question. However, the formulas we align in our paper originate from [1], with the goal of learning a good density. Specifically, our Equation (5) updates the VAE objective using the ELBO loss, aligning with Equation (7) in [1]. Our Equation (6) calculates the density probability using importance sampling, aligning with equation (8) in [1].
>
> The Euqation (4) in CLUE stems from Euqation (7) in [1], and Euqation (5) in CLUE is an approximation of Euqation (4). In [1], the role of importance sampling is to estimate a more accurate log probability than the ELBO, which is a lower bound for log density. Therefore, we adopt importance sampling in our Euqation (6). Meanwhile, please distinguish between CLUE and [1]: CLUE aims to learn a better representation, while [1] aims to learn a better density.
>
> **re (Q.1.2)** Our motivation for choosing VAEs is that past work [1] has demonstrated their powerful effectiveness in representing density-defined supports. We do not directly leverage the latent space information, rather, our paper primarily utilizes the density estimation capability of VAEs. This is because logP(a∣s) mathematically corresponds to log density, and thus we require a robust density estimator. Both $P^{∗}(a∣s)$ and $\hat P(a∣s)$ are computed via importance sampling, and [1] has claimed that the ELBO (Empirical Lower Bound) is a lower bound for log density, while inferencing based on importance sampling provides a closer approximation to the log density.
>
> **Q.2** Regarding theorem 4.2 (theorem D.1 in the appendix). Can you clarify the last move in Eq 18 in the appendix, which probabilistic model do you assume the policy follows?
>
> **re (Q.2)** Thanks for this important question. This modeling approach is more commonly seen in Gaussian policies, which are suitable for continuous decision-making problems. For discrete decision-making problems, the MSE (Mean Squared Error) can be replaced with cross-entropy.
>
> # Reference
>
> [1] Jialong Wu, Haixu Wu, Zihan Qiu, Jianmin Wang, Mingsheng Long, Supported Policy Optimization for Offline Reinforcement Learning

---

> ### Author Response · Authors · 2024-11-20
> **Reply to Reviewer eWnu's followed question regarding wk 1.1**
>
> Thanks for your question. It is very unlikely that the situation where $\log \frac{\hat P}{P^*} = 0$ will occur.
>
> The reason is that *VAE is used as a density estimator. This implies that the probability of any given sample is also related to the data used. Meanwhile, since the demo is only a part of the expert dataset, it is difficult for* $P^*(a|s)$ *to equal* $\hat{P}(a|s)$ *for any given* $(s, a)$.

---

> ### Author Response · Authors · 2024-11-25
> **Dear Reviewer eWnu, if you have any unresolved questions, please feel free to add comments.  Thank you!**
>
> If you have any unresolved questions, please feel free to add comments.  Thank you!

---

> ### Author Response · Authors · 2024-11-25
> **Further Reply to Reviewer eWnu's followed question regarding wk 1.1**
>
> To further address the your concerns, we support our previous response *The reason is that VAE is used as a density estimator. This implies that the probability of any given sample is also related to the data used. Meanwhile, since the demo is only a part of the expert dataset, it is difficult for* $P^*(a|s)$ *to equal* $\hat{P}(a|s)$ *for any given* $(s, a).$ through a simple experimental comparison.
>
> Specifically, to support our analysis, we further conduct a comparison through an experimental result. We test ADR separately on medium-expert and medium, where the dataset in medium-expert contains trajectories of both medium and expert types.
>
> Specifically, we can observe that the performance of ADR on medium-expert is better than that on medium. Therefore, even if the trajectories sampled from $\hat D$ are close to the demo trajectories, it is unlikely to encounter the situation where $\log\frac{\hat P}{P^*}=0$. Otherwise, the performance of ADR on medium-expert and medium should be the same.
>
> |medium-expert|score|medium|score|
> |--|--|--|--|
> |hopper-me |109.1$\pm$3.2 |hopper-m |69.0$\pm$1.1|
> |halfcheetah-me| 74.3$\pm$2.1 |halfcheetah-m| 44.0$\pm$0.1|
> |walker2d-me |110.1$\pm$0.2 |walker2d-m |86.3$\pm$1.7|
> |Ant-me| 132.7$\pm$0.3 |Ant-m |106.6$\pm$0.5|
> |Avg.| 106.6 |Avg. |76.5|
>
> We kindly invite you to further consider our follow-up response. Thank you very much!

---

> ### Author Response · Authors · 2024-12-01
> **Official Comments by Author**
>
> We thanks for the reviewer's response, but as for what constitutes theory, we do not agree that finding an intuitively appealing example, as you suggested, qualifies as theory. Theory should encompass proofs and analyses, which are the elements present in the ADR paper. If your concern is solely about the validity of our assumptions, please raise specific questions regarding them. I do not believe that reviewing based solely on intuition is a good approach.
>
> Here, we request Area Chair (AC) to thoroughly read our dialogue.

---

> > ### Author Response · Authors · 2024-12-01
> > **In terms of wheather  $\log \frac{\hat P}{P^*} = 0$ will occur.**
> >
> > *At the same time, we have already explained to you why your thoughts are incorrect:*
> >
> > Thanks for your question. It is very unlikely that the situation where $\log \frac{\hat P}{P^*} = 0$ will occur.
> >
> > The reason is that *VAE is used as a density estimator. This implies that the probability of any given sample is also related to the data used. Meanwhile, since the demo is only a part of the expert dataset, it is difficult for* $P^*(a|s)$ *to equal* $\hat{P}(a|s)$ *for any given* $(s, a)$.

---

> > ### Comment · Reviewer_eWnu · 2024-12-01
> > **A reply to the authors**
> >
> > In my review, I raised concerns regarding the theoretical justification of ADR. I don't think you managed to solve this issue in your response.
> >
> > Regarding toy examples, they can indicate patterns that lead to reasoning, and I do think they bear importance in strengthening your argument. That said, I don't think they should come on their own and they should be accompanied by theory and practical evaluations.
> >
> > About the case where P*=P^ I don't think your argument is strong, as you rely on practical learning mechanisms to avoid the theoretical problem raised. In any case when P^ -> P* the distance between ADR and P* grows (as also indicated by your theoretical analysis).
> >
> > I encourage the AC as well as other reviewers to carefully read my review, which includes additional concerns, e.g. regarding other aspects of ADR like the justification of the ADE loss, and see whether the authors successfully addressed these concerns.

---

> > > ### Author Response · Authors · 2024-12-01
> > > **Thank you very much for inviting AC to assess whether I have solved your problem.**
> > >
> > > *I am still highlighting my previous response separately:*
> > >
> > > Thanks for your question. It is very unlikely that the situation where $\log \frac{\hat P}{P^*} = 0$ will occur.
> > >
> > > The reason is that *VAE is used as a density estimator. This implies that the probability of any given sample is also related to the data used. Meanwhile, since the demo is only a part of the expert dataset, it is difficult for* $P^*(a|s)$ *to equal* $\hat{P}(a|s)$ *for any given* $(s, a)$.

---

> > > > ### Comment · Reviewer_eWnu · 2024-12-01
> > > > **regarding P*=P^**
> > > >
> > > > Notice that you don't need that P* equals P^ for all *states* to obtain this peculiarity.
> > > > For any given state if P*=P^ the objective function for this state vanishes.
> > > >
> > > > When representing pi with a neural network, the output is obviously correlated to similar states so this issue can be sometimes obfuscated by practical learning considerations, but nonetheless, it exists and prevents ADR from converging to P*

---

> ### Author Response · Authors · 2024-12-01
> **replay: regarding $P^*=\hat P$**
>
> Here, we will summarize responses related to the **no meaningful**  $P^*=\hat P$
>
> 1. The first is through analysis. The demo only contains one trajectory, while the suboptimal dataset has far more than that (over 1000). If the suboptimal dataset differs significantly from the demo, their densities are unlikely to be the same. Even if the suboptimal pose dataset fragements were the same as the demo, their densities would still differ due to different data distributions.
>
> 2. The second is through experimentation. I presented experiments comparing demo to medium-expert and demo to medium. The experimental results indicate that the former outperforms the latter. If your conclusion is correct, then the performance of the two should be the same.
>
> |medium-expert|score|medium|score|
> |--|--|--|--|
> |hopper-me |109.1$\pm$3.2 |hopper-m |69.0$\pm$1.1|
> |halfcheetah-me| 74.3$\pm$2.1 |halfcheetah-m| 44.0$\pm$0.1|
> |walker2d-me |110.1$\pm$0.2 |walker2d-m |86.3$\pm$1.7|
> |Ant-me| 132.7$\pm$0.3 |Ant-m |106.6$\pm$0.5|
> |Avg.| 106.6 |Avg. |76.5|

---

> > ### Author Response · Authors · 2024-12-01
> > **Regarding the rebuttal to  Reviewer eWnu**
> >
> > Dear  Reviewer eWnu
> >
> > We appreciate the valuable time you took to write your comments during the review process, even though we do not agree with some of your points. Additionally, the questions raised by other reviewers have helped us improve our paper, and we have made some revisions accordingly. Therefore, we are not rebutting your questions without purpose, we just believe that some of your questions are not appropriate.
> >
> > Thanks for your efforts again.
> >
> > Sincerely,
> >
> > Conference Authors.

---

### Official Review · Reviewer_7oCa · 2024-10-31

**Soundness:** 3
**Presentation:** 3
**Contribution:** 3
**Rating:** 6
**Confidence:** 4

**Summary:**

The author propose a simple yet effective one-step supervised IL framework termed Adversarial Density Regression (ADR), which leverages demonstrations to correct the policy distribution learned from datasets of unknown-quality toward expert distribution without relying on the Bellman operator.

**Strengths:**

The author introduces a substantial and well-validated advancement in IL with Adversarial Density Regression (ADR). ADR addresses key limitations in current IL methodologies by diverging from the typical reliance on multi-step Bellman updates and conservative RL policies, which can introduce cumulative errors and struggle with out-of-distribution (OOD) data. Experimentally, ADR surpass selected IL algorithms and achieves better performance than the offline RL algorithm IQL in the Android domain.

**Weaknesses:**

This paper introduces ADR and demonstrates strong results in the Gym-Mujoco, Kitchen, and Adroit domains. However, these environments primarily focus on continuous control tasks with static distributions. This choice does not fully capture the complexity of real-world applications, which often involve changing dynamics or high-dimensional, sparse observation spaces (e.g., partially observable or dynamic obstacle environments). Testing ADR on environments with dynamically shifting conditions, like robotic control with dynamic obstacles or changing goals, could provide more insights into its robustness and generalizability. ADR's density-weighted objective, VAE-based density estimation, and adversarial learning framework are likely to introduce significant computational overhead, this paper limited analysis of computational efficiency.

**Questions:**

Could the authors comment on how ADR would handle more substantial or structured noise?
Could the authors provide insights into ADR’s computational demands and compare them to other IL methods?
Could the authors clarify the benefits of using adversarial learning here? For instance, how much does Adversarial Density
Estimation (ADE) contribute to ADR’s overall performance?

---

> ### Author Response · Authors · 2024-11-21
> **Reply to Reviewer 7oCa's question section (part 1)**
>
> **Q.1** Could the authors comment on how ADR would handle more substantial or structured noise?
>
> **re (Q.1)** Thanks for your valuable question. As shown in Figure 4, we tested the robustness of ADR to noise under different Gaussian noise weights. The experimental results indicate that ADR does not show a significant performance decline on tasks other than ant. This is because ADR is a density-weighted BC objective, and it is aided by a large amount of suboptimal data during training, making ADR relatively stable.
>
> Meanwhile, we propose a solution to mitigate the impact of increased noise in demos by reducing the number of demos. The rationale behind this approach is that noise has a greater impact on experimental settings with more demos, as more demos mean a higher percentage of training samples will be used by the model for training, thereby directly affecting the policy. Additionally, ADR can achieve considerable performance without requiring too many demos. Specifically, as shown in Table 2, our tests on the androit and kitchen domains were conducted with only one expert trajectory as the demo. As shown in Table 3, our tests on the gym-mujoco domain were conducted with five expert trajectories as the demos.
>
> To validate our idea, we trained with different weights under LfD (20) and LfD (5). We found that LfD (20) is more severely affected by noise compared to LfD (5), especially on the ant task. Correspondingly, the results of LfD (5) are relatively stable, with no significant performance decline. Notably, the noise weight set for LfD (5) is greater than that for LfD (20).
>
> |LfD (20)|$w$=0|$w$=0.3|$w$=0.9|
> |:--:|--|--|--|
> |hopper-m|69.0$\pm$1.1|68.5$\pm$0.6|68.9$\pm$2.0|
> |walker2d-m|87.9$\pm$0.7|82.3$\pm$0.2|82.0$\pm$0.7|
> |Ant-m|106.6$\pm$0.5|103.5$\pm$0.5|103.5$\pm$1.0|
> |halfcheetah-m|44.0$\pm$0.1 |43.8$\pm$0.2|43.5$\pm$0.2|
>
> |LfD (5)|$w$=0|$w$=0.5|$w$=1.5|
> |:--:|--|--|--|
> |hopper-m|69.0$\pm$1.1|65.9$\pm$0.2|68.5$\pm$2.4|
> |walker2d-m|86.3$\pm$1.7|82.9$\pm$1.2|81.7$\pm$0.2|
> |Ant-m|106.6$\pm$0.5|104.6$\pm$0.1|104.1$\pm$0.8|
> |halfcheetah-m|44.0$\pm$0.1 |43.6$\pm$0.1|43.2$\pm$0.1|
>
> **Q.2.2** Could the authors provide insights into ADR’s computational demands and compare them to other IL methods? **Q.2.1**Could the authors clarify the benefits of using adversarial learning here?
>
> **re (Q.2.1)** Apart from the need to pretrain two VAEs, ADR does not require significant computational resources, as it is a single-step density-weighted BC objective that only requires supervised training based on Equation 10 on an offline dataset. In contrast, imitation learning algorithms such as IQ-Learn or GAIL, e.g. first need to estimate a Q-function or a reward function, and then learn the Q-function based on the reward, followed by updating the policy through an Actor-Critic approach. The process of learning the Q-function and the policy alternates, resulting in a time complexity that is at least two to three times higher than that of ADR.
>
> **re (Q.2.2)** Thanks for your insightful question. We will explain our method from the perspective of adversarial learning. First, let us derive from Equation 1:
>
> $
> E_{\mathcal{D}}[D_{KL}(\pi||\pi^*)-D_{KL}(\pi||\hat \pi)] = E_{\mathcal{D}}[\pi*(\log \hat \pi -log \pi^*)] \\
> $
>
> By observing the objective function on the right-hand side of this equation, we encounter two scenarios: when $D$ is closer to $D^*$, the dataset sampled from the expert policy, $E_{D}[(\log \hat \pi - \log \pi^*)] > 0$, conversely, when $D$ is closer to $\hat D$, the dataset sampled from a suboptimal policy, $E_{D}[(\log \hat \pi - \log \pi^*)] < 0$. Therefore, maximizing the likelihood of trajectory segments that are closer to the expert policy and minimizing the likelihood of non-expert trajectories achieves the effect of screening out a dataset in $\hat D$ that is closer to $D^*$, thereby augmenting the data.

---

> > ### Author Response · Authors · 2024-11-21
> > **Reply to Reviewer 7oCa's question section (part 2)**
> >
> > **Q.3** For instance, how much does Adversarial Density Estimation (ADE) contribute to ADR’s overall performance?
> >
> > **re (Q.3)** Thanks for your questions. In the ablation study section, we have conducted different tests on each component of ADR. Specifically, we have labeled ADR without ADE this setting as "wo-ADE" in Figure 3.
> >
> > Meanwhile, in this paper, we have adopted relatively novel aggregated RL metrics to evaluate how much confidence will decrease when the ADE term is removed from the VAE's objective function (i.e., "λ=0") compared to the original ADR objective function. Specifically, in Figure 3(a), the experimental results of the "wo-ADE" setting show a probability of over 50% but less than 55% to outperform ADR across all selected gym-mujoco tasks. Therefore, ADE has an impact on the performance of ADR. Additionally, we have compared each task in Figure 3(b) and found that the algorithm with ADE does not perform well on the medium-replay dataset, with a confidence level close to 90% or above. Meanwhile, the algorithm without ADE performs better on the medium and medium-expert datasets. Thus, ADE may provide more benefit to the algorithm on the medium/medium-expert datasets than on the medium-replay dataset.

---

> > > ### Comment · Reviewer_7oCa · 2024-11-22
> > > **Thanks for your response!**
> > >
> > > Thank you for addressing my questions. I have no further concerns and keep the 6 rating.

---

> > > > ### Author Response · Authors · 2024-11-22
> > > > **Thanks for your reply!**
> > > >
> > > > We appreciate your insightful and valuable questions!

---

### Official Review · Reviewer_jPp9 · 2024-11-04

**Soundness:** 2
**Presentation:** 3
**Contribution:** 2
**Rating:** 5
**Confidence:** 4

**Summary:**

This paper introduces a one-step supervised imitation learning (IL) framework that addresses the challenge of training policies on suboptimal datasets while aligning them with expert demonstrations. The proposed method leverages two variational auto-encoders (VAEs), trained separately on the suboptimal data and expert demonstrations, to estimate the respective behavior distributions. These VAEs are then employed to calculate density weights for offline policy training. This training is performed using an adversarial density regression (ADR) approach. ADR minimizes the Kullback-Leibler (KL) divergence between the policy’s behavior distribution and the expert demonstration distribution while simultaneously maximizing the KL divergence between the policy and the suboptimal dataset distribution. This formulation translates into a density-weighted regression between the policy’s output and the actions observed in the suboptimal dataset. Notably, this approach circumvents the reliance on the Bellman operator and demonstrates competitive performance compared to other offline reinforcement learning (RL) methods. The proposed framework exhibits robustness to noisy demonstrations and mitigates the risk of out-of-distribution (OOD) generalization issues.

**Strengths:**

1. **Empirical Validation**: The paper demonstrates the efficacy of the proposed Adversarial Density Regression (ADR) method through comprehensive experimental results. ADR consistently outperforms baseline algorithms across diverse task domains, achieving higher accuracy while exhibiting robustness to noisy demonstrations and a reduced risk of out-of-distribution generalization.
2. **Training Efficiency**: The one-step supervised learning paradigm employed by ADR effectively minimizes cumulative errors, resulting in stable training without tuning conservative terms. This streamlined approach contributes to the method’s practical appeal.
3. **Theoretical Analysis**: The authors provide theoretical for ADR’s effectiveness. They prove that minimizing the ADR objective function is aligned with attaining an optimal policy, which forms a valid theoretical justification for the proposed method.

**Weaknesses:**

1. **Novelty**: The main innovation in Density Weighted Regression (DWR) is the importance sampling term combined with a behavior cloning objective. However, the novelty is unclear. The justification relies on comparisons to traditional Behavior Cloning (BC) and DICE, but the connection to DICE seems weak. DICE addresses a reinforcement learning problem using its dual form, involving a Bellman flow constraint, while ADR focuses on an imitation learning problem. These approaches are based on different formulations.
2.  **Computational Overhead**: The reliance on auxiliary Variational Autoencoder (VAE) training for each task raises concerns regarding computational efficiency. Although the paper does not explicitly address the associated computational cost, this aspect warrants further investigation.
3. **Technical Significance and Soundness.**
- Theorem 4.2 seems to be an obvious reformulation of equation (8). The necessity of naming it as a theorem is unclear. For example, what's the main observation and conclusion from this theorem is unknown.
- The application of KL-divergence has several limitations. For instance, it requires that the compared distributions have exactly the same support, and it isn't a true distance metric because it's not symmetric. Nowadays, more promising alternatives like Wasserstein distances are gaining attention, with wide exploration in reinforcement learning applications.

**Questions:**

1. Dataset Size: The paper could benefit from an analysis of the impact of the demonstration dataset size on ADR’s performance. Could you please demonstrate the relationship between dataset size and model accuracy?
2. Noisy Data Handling: It would be beneficial to investigate the robustness of ADR in the presence of noisy or low-quality suboptimal datasets. Could you please demonstrate the capability of your method to address noisy or low-quality suboptimal data issue?
3. VAE Choice: A clear justification for selecting VAEs as the behavior distribution estimator would enhance the paper’s clarity. Could you please discuss the advantages of VAEs compared to other potential estimators?

---

> ### Author Response · Authors · 2024-11-15
> **The reason for the early reply to Reviewer jPp9's Weaknesses' section**
>
> Thanks for your questions and suggestions. And we appreciate your initial recognition of our theoretical analysis and experimental results. We have carefully read your comments provided. Upon initial assessment, we believe that some of your questions may require extensive communication. Meanwhile, given that reviewers are generally quite busy, we have decided to submit an early response to address some of the your theoretical concerns, such as the relationship between this terminology'DICE' and our method e.g.
>
> For some questions sourced from question section, such as those related to different density estimators, we will provide analysis, and we will appropriately supplement experiments latter to further analyze these aspects from an experimental perspective. (We will respond as soon as possible during the rebuttal period)
>
> Thank you

---

> > ### Comment · Reviewer_jPp9 · 2024-11-22
> > **Thanks for your response**
> >
> > I would like to thank the author for their response. It seems the authors are starting to question whether DICE is a proper wording in their title. I am confused by the statement that "DICE-style" is not equivalent to "DICE". In my mind, this is an imitation learning work. The difference to DICE is apparent. From the perspective of IL, ADR is a resampling extension of IL (see equations (9) and (10)). The novelty is a major concern. To improve the paper, the author might consider renaming their paper and reformulating their contribution.

---

> ### Author Response · Authors · 2024-11-15
> **Reply to Reviewer jPp9's Weaknesses' section**
>
> **wk.1** Novelty: The main innovation in Density Weighted Regression (DWR) is the importance sampling term combined with a behavior cloning objective. However, 1.2) the novelty is unclear. 1.1) The justification relies on comparisons to traditional Behavior Cloning (BC) and DICE, but the connection to DICE seems weak. DICE addresses a reinforcement learning problem using its dual form, involving a Bellman flow constraint, while ADR focuses on an imitation learning problem. These approaches are based on different formulations.
>
> **re (wk.1.1)** In this paper, we have not referred to our algorithm as a derivative of DICE. Instead, we have termed our algorithm as a DICE-type (in line 20) supervised learning algorithm. Specifically,
>
>   **Why utilizing the word 'DICE type' in lines 20.** Our motivation for considering ADR as a DICE-type algorithm (in line 20) is:
> ADR also leverages a demo dataset to refine the distribution learned by the policy on other datasets, which shares a similarity in the 'regularization' with the DICE algorithm in that it incorporates a distance regularization term in the reward function to adjust the learned policy.
>
>   In addition, due to the limited word count in the abstract, we prefer to utilize concise language to summarize the main content of this paper.
>
>   **Why mentioned DICE in our paper.** Furthermore, the reason we chose DICE for comparison (lines 51~55. On the other hand...), is that DICE algorithms also use the experimental setup of "demo + other data" to train a policy that is close to the demo policy. This experimental setup is relatively close to our experimental setup. Meanwhile, some literatures [1] written from IL, and these literatures are very solid.
>
>   **DICE is RL or IL?** During our investigation of DICE-related literature, we encountered some ambiguities, as the experimental setup of DICE is close to imitation learning (IL) setups, and some literature initializes the introduction from an IL perspective [1]. However, some DICE algorithms use reward functions, which has a strong connection with reinforcement learning (RL). Therefore, in our paper, we use the term "another approach" (lines 52) instead of categorizing DICE definitively under IL, however, there having similar settings between DICE and IL. Correspondingly, In the preliminary section, we introduce IL and DICE separately.
>
>   It is also important to note that we are aware of some DICE algorithms involve modifications to the reward function, which differ significantly from our method. Furthermore, most DICE papesr conduct theoretical analysis from the perspective of occupancy, while our paper conducts theoretical analysis from the perspective of return boundary. Therefore, we choose to use the term "DICE-type" rather than "DICE".
>
>   Given the historical context of DICE-related literature, we believe that our current writing balances the relationship beween RL/RL/IL and DICE from both the literature and experimental setup perspectives. Additionally, we have included supplementary explanations on this matter in an Appendix page of the latest version of our manuscript.
>
> **re (wk.1.2)** At the beginning of this paper, we mentioned that our algorithm falls under the category of imitation learning algorithms, given this context.
> - Our method belongs to an imitation learning (IL) algorithm. During the update process, we do not use the Bellman operator, thus our method does not suffer from offsets accumulated under suboptimal rewards, which we believe is a crucial point. Additionally, some research [2] has also discussed that the reward shaping methods of some previous IL algorithms do not guarantee 100% optimal rewards.
> - Meanwhile, our method does not include a conservative term, which results in a lower dependence on parameters for each task when testing offline tasks. Specifically, our algorithm completed all experiments using only few sets of parameters. As we all know, RL algorithms are sensitive to parameters, and most IL algorithms update their policies based on the RL framework and the reward function obtained through imitation learning. Therefore, there are parameter tuning issues stemming from the RL framework.
> - Furthermore, our algorithm demonstrates potential in terms of effectiveness within the domain of simulation environments.
>
> In the paper, we mentioned algorithms related to DICE and discussed them in the introduction. Furthermore, given that the term "DICE-type" has caused some confusion for the reviewers in understanding the paper, we are considering whether to make necessary modifications to this term, and we will provided additional explanations on the supplementary page latter. If the reviewers strongly recommend modifying this term, we will describe the learning mechanism of ADR in the abstract section using other more appropriate terms.

---

> ### Author Response · Authors · 2024-11-15
> **Reply to Reviewer jPp9's Weaknesses' section (part 2)**
>
> **wk.2** The reliance on auxiliary Variational Autoencoder (VAE) training for each task raises concerns regarding computational efficiency. Although the paper does not explicitly address the associated computational cost, this aspect warrants further investigation.
>
> - wk.2.1 1) We have optimized the VAE weighted objective function. Specifically, assuming the input matrix has dimensions of $b\times m$, we simplified the objective function from the original objective $min_{\pi} E_{D}[D_{KL}(\pi||P^*)-D_{KL}(\pi||\hat P)]$(Equation 9) to $min_{\pi}E_{D}[\log\frac{\hat P}{P^*} ||\pi(\cdot|s)-a||^2]$ (Equation 10), thereby reducing the complexity from O(2(1+b)b*m) to O(b*m). We have compared the two in the supplementary page (Figure 6: (Left)), and it can be seen that Equation (10) requires less time compared to Equation (9) under the same training settings.
> - wk.2.2 Common density estimators include Gaussian, VAE (Variational Autoencoder), transformer, and diffusion-based density estimators. While diffusion models have stronger representation capabilities, they require multi-step sampling during the sampling process, resulting in higher time complexity compared to VAE and Gaussian models.
>
>
>
>   Furthermore, considering this issue from the perspective of reinforcement learning (RL), although Gaussian may require fewer parameters, VAE is more suitable than Gaussian as a density estimator in RL [3]. Therefore, our method performs well and has lower computational complexity compared to diffusion models. Additionally, our research is conducted in the context of Markov Decision Processes (MDPs), while diffusion models and transformers are generally sequence models used in non-MDP settings, which is hard to be implemented under MDP settings. For exmaple, given current examples {$s_0,a_0$}, we can't find {$s_{0-k},a_{0-k},\cdots s_0,a_0$.} for sequence models.
>
> **wk.3.1** Theorem 4.2 seems to be an obvious reformulation of equation (8). The necessity of naming it as a theorem is unclear. For example, what's the main observation and conclusion from this theorem is unknown. **wk.3.2** The application of KL-divergence has several limitations. For instance, it requires that the compared distributions have exactly the same support, and it isn't a true distance metric because it's not symmetric. Nowadays, more promising alternatives like Wasserstein distances are gaining attention, with wide exploration in reinforcement learning applications.
>
> **re (wk.3.1)** Thanks for your suggestion. We have renamed Theorem 4.2 from "density weight" to "weighted linear objective can implement adversarial density regression." We have chosen to name this conclusion Theorem 4.2 because the objective function in Theorem 4.2 (Equation 10) can significantly improve the training efficiency of Equation (9). Additionally, in the Appendix page Figure 6 (Left), we have compared Equation (10) and Equation (9) under the same experimental setup, and the results show that the formula in Theorem 4.2 performs more efficient. Therefore, we need a conclusion to justify the rationality of the high correlation between our Equations (10) and (9), ensuring that the theoretical motivation of our method is reasonable and that the real implementation is well-founded.
>
>
>
> **wk.3.2** The application of KL-divergence has several limitations. For instance, it requires that the compared distributions have exactly the same support, and it isn't a true distance metric because it's not symmetric. Nowadays, more promising alternatives like Wasserstein distances are gaining attention, with wide exploration in reinforcement learning applications.
>
> **re (wk.3.2)** We acknowledge the limitations of KL divergence. However, we derive the KL-based objective into a density-weighted objective (Equation 10). In Equation 10, we base our approach on the density-weighted BC objective, which is inherently symmetric because ||a-b|| = ||b-a||.
> In addition, we compared our method with the Optimal Transport-based approach [4], and the experimental results showed that our method has advantages on long-distance tasks such as Andtroit (*door-cloned , door-human, hammer-cloned, hammer-human,pen-cloned,pen-human, relocate-cloned, relocate-human*) and Kitchen (*kitchen-mixed, kitchen-partial, kitchen-completed*) in our Table 2. We list the results for each domain here. In particular, oracle denotes the GT sparse (GT) reward.
> |IL Tasks (LfD (1)) |BC |IQL (oracle) |IQL (OTR) |ADR|
> |--|--|--|--|--|
> |Total (Androit) |104.5| 118.1| 122.2| **263.5**|
> |Total (Kitchen&Androit) |259| 277.9| 272.2 | **526.6**|
>
>
> # Reference
>
> [1] Jason Ma, etal. Versatile Offline Imitation from Observations and Examples via Regularized State-Occupancy Matching
>
> [2] anonymous2024trajectorylevel,Trajectory-level Data Generation with Better Alignment for Offline Imitation Learning
>
> [3] Wu, etal. Supported Policy Optimization for Offline Reinforcement Learning
>
> [4] Luo,etal. OPTIMAL TRANSPORT FOR OFFLINE IMITATION LEARNING

---

> ### Author Response · Authors · 2024-11-21
> **Reply to Reviewer jPp9's questions' section**
>
> **Q.1** Dataset Size: The paper could benefit from an analysis of the impact of the demonstration dataset size on ADR’s performance. Could you please demonstrate the relationship between dataset size and model accuracy?
>
> **re (Q.1)** Thanks for your questions. We selected the medium dataset for testing and adjusted the number of demonstrations accordingly. We found that different tasks have varying requirements for the number of demonstrations. Specifically, based on the experimental results provided, we observed that Ant and Hopper can benefit from increasing the number of demonstrations, with Ant showing a particularly significant improvement of nearly 30 points. However, Walker and HalfCheetah showed less improvement. We believe that the more significant improvement on Ant is due to the higher degree of freedom in the Ant environment and the diversity of the dataset. Therefore, when using ADR, it is easier to leverage demonstrations to capture trajectory segments close to expert performance in suboptimal datasets, thereby achieving performance close to that of experts.
>
> |task name|n (demo)=5|n (demo)=10|n (demo)=20|
> |:--:|--|--|--|
> |hopper-m|69.0$\pm$1.1|69.9$\pm$1.9|**71.0**$\pm$1.5|
> |walker2d-m|**87.9**$\pm$0.7|86.7$\pm$1.7|86.3$\pm$1.7|
> |Ant-m|106.6$\pm$0.5|108.9$\pm$1.9|**133.6$\pm$0.4**|
> |halfcheetah-m| 44.0$\pm$0.1|44.0$\pm$0.1|44.0$\pm$0.1|
> |Avg.| 76.875|77.375|**83.725**|
>
> **Q.2.2** Noisy Data Handling: It would be beneficial to investigate the robustness of ADR in the presence of noisy or low-quality suboptimal datasets. **Q.2.1** Could you please demonstrate the capability of your method to address noisy or low-quality suboptimal data issue?
>
> **re (Q.2.1)** ADR still demonstrates outstanding performance on low-quality datasets. According to the description in D4RL [5], medium-replay is the replay buffer from SAC training when it reaches half of the expert's performance. Its overall quality and quantity are less than those of the medium dataset. We found that ADR outperforms previous major imitation learning and DICE methods.
>
> |LfD (5) |ORIL (TD3+BC)| SQIL (TD3+BC) |IQ-Learn| ValueDICE| DemoDICE |SMODICE| ADR|
> |--|--|--|--|--|--|--|--|
> |hopper-mr |26.7| 27.4 |15.4| 80.1| 26.5| 34.9 | **74.7**±1.7|
> |halfcheetah-mr |2.7 |15.7| 4.8 |0.9 |38.7| 38.4 |**39.2**±0.1
> |walker2d-mr |22.9| 7.2| 10.6 |0 |38.8| 40.6| **67.3**±4.7|
> |Ant-mr| 24.5 |23.6| 27.2| 32.7| 68.8| 69.7| **95.4**±1.1|
> |Avg.|19.2|18.5|14.5|28.4|43.2|45.9|**69.2**|
>
> **re (Q.2.2)** ADR is highly robust to noise in demonstrations. In Figure 4 of this paper, we demonstrate the effect of ADR when varying levels of Gaussian noise are added to the demonstrations. Specifically, in Figure 4, we selected the medium dataset as the suboptimal data, and we found that only Ant was significantly affected. ADR was sufficiently robust to noise in other tasks. We believe that ADR's high robustness to noise is due to its weighted BC objective, which provides greater stability. In contrast, the reward shaping approach learns a suboptimal reward, which can lead to cumulative offsets when further learning the Q-function, thereby affecting the policy's performance.
>
> **Q.3** VAE Choice: A clear justification for selecting VAEs as the behavior distribution estimator would enhance the paper’s clarity. Could you please discuss the advantages of VAEs compared to other potential estimators?
>
> **re (Q.3)** Common density estimators include Gaussian, VAE (Variational Autoencoder), transformer, and diffusion-based density estimators.
>
> - diffusion: While diffusion models have stronger representation capabilities, they require multi-step sampling during the sampling process, resulting in higher time complexity compared to VAE and Gaussian models. dditionally, our research is conducted in the context of Markov Decision Processes (MDPs), while diffusion models and transformers are generally sequence models used in non-MDP settings, which is hard to be implemented under MDP settings. For exmaple, given current examples {$s_0,a_0$}, we can't find {$s_{0-k},a_{0-k},\cdots s_0,a_0$.} for sequence models.
>
> - Gaussian: Although Gaussian may require fewer parameters, Gaussian cannot fit complex nonlinear distributions. And, VAE is more suitable than Gaussian as a density estimator in RL [3].
>
> In summary, compared to diffusion models, VAEs are more suitable for our problem setting, meanwhile, VAE has lower computational complexity compared to diffusion models. Additionally, compared to Gaussian models, VAEs yield better results. Therefore, VAEs balance the effectiveness of density estimation and computational efficiency, making them a good choice for our problem.
>
> # Reference
>
> [5] Justin Fu, Aviral Kumar, Ofir Nachum, George Tucker, Sergey Levine. D4RL: Datasets for Deep Data-Driven Reinforcement Learning

---

> ### Author Response · Authors · 2024-11-22
> **Further reply to Reviewer jPp9's official comment**
>
> Thanks for your reply.
>
> **Q.1** I am confused by the statement that "DICE-style" is not equivalent to "DICE"
>
> **re (Q.1)** Thanks for your suggestions. We agree with your viewpoint, and we did claim at the beginning of the paper (lines 11~12) that this is a one-step supervised imitation learning framework. Additionally, we did not categorize DICE as an imitation learning algorithm in the paper, as we have also mentioned in our previous response to you.
>
> **In order to resolve the confusion caused by the term DICE-type for the reviewers in understanding the paper, we plan to replace the term "DICE-type" with "imitation learning with auxiliary imperfect demonstration." This description has also received support from reviewer 8KXJ.**
>
> *We will make the necessary revisions in the manuscript and will indicate the locations of the changes in our public official comments once the revisions are complete.*
>
> **Q.2** From the perspective of IL, ADR is a resampling extension of IL (see equations (9) and (10)). The novelty is a major concern. To improve the paper, the author might consider renaming their paper and reformulating their contribution.
>
> **re (Q.2)** Thanks for the your consideration of the ADR contribution. As mentioned in the paper (lines 73-76), ADR is a fully supervised imitation learning algorithm that does not rely on the Bellman operator. To our knowledge, most imitation learning algorithms are based on the Bellman operator and involve reward shaping and Q-estimating processes. Additionally, under suboptimal rewards, these offline paradigms based on the Bellman operator can learn accumulated offsets. However, ADR is a single-step supervised learning paradigm, so it is not affected by such accumulated offsets. Furthermore, as mentioned in the paper (lines 77-83), ADR has theoretical proof that its single-step update optimization objective is consistent with policy convergence to the optimal policy. Therefore, this novel training paradigm of ADR also has theoretical guarantees.
>
> **In particular**, we would like to further clarify the distinction between multi-step and single-step methods here. As illustrated in Figure 2, the multi-step approach based on the Bellman operator involves an alternating learning process between Q-values and policies. In contrast, the ADR-based approach is a single-step learning process, where each computation brings us closer to the optimal policy.
>
> Therefore, whether from the perspective of the training paradigm or the theoretical level, ADR is not a simple resampling of previous algorithms, but a novel single-step imitation learning (IL) paradigm.
>
> Regarding DICE, since we did not categorize DICE as IL, even if there are differences between DICE and IL in the reviewer's opinion, this will not affect our innovation.

---

> > ### Comment · Reviewer_jPp9 · 2024-11-22
> > **Further concern**
> >
> > The claim that "imitation learning algorithms are based on the Bellman operator and involve reward shaping and Q-estimating processes" is incorrect. Imitation learning is a classic method that is based on supervised learning. It was not until recent years that people started to combine RL with Imitation, creating well-known algorithms like GAIL and IRL. Traditional algorithms like Dagger and BC are all based on supervised learning.

---

> > > ### Author Response · Authors · 2024-11-22
> > > **Further reply to Reviewer jPp9's Further concern**
> > >
> > > Thanks very much for your furhter reply.
> > >
> > > We appreciate the your further correction of this claim. In our contribution, we will confine our innovation within the category of RL-combined imitation learning algorithms, as our approach is still analyzing the convergence range of returns, thus making it comparable to a certain extent.
> > >
> > > Regarding BC, although this algorithm was proposed a long time ago and BC falls into the category of supervised learning-based IL, we mention in line 132 of the paper: "however, BC's performance is brittle when $D^*$ is scarce." Furthermore, as seen in Tables 2 and 3, ADR demonstrates significant advantages over BC on Androit, Kitchen (LfD (1)), and gym-mujoco (LfD(5)). Additionally, both DWR and ADE, mentioned in our article, can be used to address this limitation. Our ablation experiments also confirm the effectiveness of ADE and DWR. Therefore, ADR and BC are two different approaches.
> > >
> > > Meanwhile, Dagger is an online algorithm whose goal is to continuously augment the dataset through online interaction to enhance policy performance. In contrast, our algorithm aims to learn a good policy from a static dataset, which is closer to the paradigm of offline learning.
> > >
> > > Therefore, We intend to describe our contribution as:
> > >
> > > *Unlike most modern RL-combined imitation learning (IL) algorithms, which rely on the Bellman operator and incorporate reward shaping and Q-estimating processes, ADR operates as a single-step supervised learning paradigm, rendering it immune to the accumulated offsets resulting from suboptimal rewards. Meanwhile, compared to traditional single-step IL paradigms such as BC, ADR can achieve better performance with a limited number of demos based on adversarial density-weighted regression. Therefore, ADR combines the advantages of single-step updates while demonstrating superior performance compared to previous RL-combined IL approaches on the experimental level. Additionally, it has theoretical guarantees from the perspective of RL. Thus, ADR is a novel single-step supervised IL paradigm.*
> > >
> > > We will make corresponding adjustments and appropriate refinements in the text, and mark the modifications in the text with blue. Meanwhile, we will indicate the locations of the changes in our public official comments once the revisions are complete. Thanks very much for your valuable suggestions!

---

> ### Author Response · Authors · 2024-11-25
> **Dear Reviewer jPp9, if you have any unresolved questions, please feel free to add comments. Thank you!**
>
> If you have any unresolved questions, please feel free to add comments. Thank you!

---

> ### Author Response · Authors · 2024-11-30
> **Dear Reviewer jPp9,  we would like to further confirm whether we have addressed your concern, thank you.**
>
> Dear Reviewer jPp9,
>
> As the discussion deadline is approaching, we would like to further confirm whether we have addressed your concern. Thank you very much. Additionally, we humbly request that you consider increasing our score if we have successfully addressed your concerns. Thank you.
>
> Best Regard,
>
> Conference Authors

---

> > ### Comment · Reviewer_jPp9 · 2024-12-03
> >
> > Thanks for the response. In my mind, the main contribution of this paper requires some further improvement. Some misleading claims and introductions should be resolved before publication. However, in response to the author's rebuttal, it is fair to increase my rating. At this moment, I think the manuscript still remains slightly below the threshold I'd expect from papers at ICLR mostly due to the significance of the contributions and the overall clarity of presentation.

---

> ### Author Response · Authors · 2024-12-03
> **Thanks for your valuable suggestions and for improving the score**
>
> Thanks for your valuable suggestions, which have contributed to enhancing the expression of our paper, and thank you for taking the valuable time to read our comments as well as the paper. Following the your suggestions, our manuscript has better organized the relationships between DICE and other areas such as IL, and has also made some improvements in the "related work" section and others ($\textcolor{blue}{\textit{It has been marked in blue in our latest PDF}}$). Additionally, we deeply appreciate your patience in reading through our comments and for consistently providing invaluable feedback for our paper. We wish you all the best in your work and, once again, express our gratitude to you here.

---

### Author Response · Authors · 2024-11-21
**Instructions regarding the manuscript**

We are deeply grateful for the valuable suggestions provided by all the reviewers. Our manuscript have be slightly modified based on the discussions with the reviewers 8KXJ, jPp9. And the changes have be $\textcolor{blue}{\textit{highlighted in blue}}$. Thank you.

*Paper Modification*

- **(Title)** To avoid ambiguity, we have changed the paper title to ‘Imitating from auxiliary imperfect demonstrations via Adversarial Density Weighted Regression’.

- **(Abstract)** Based on the reviewers' suggestions, our paper primarily needs revisions from the perspective of presentation. Specifically, both reviewers jPp9 and 8KXJ have pointed out that the term "Dice-type" causes some interference for readers when reading this article. Since the optimization analysis method and optimization objectives of DICE differ from those of IL paradigms, the use of the term "Dice-type" should be avoided. (we replace the term "DICE-type" with the description "imitation learning with auxiliary imperfect demonstration." This description has also received support from reviewer 8KXJ)

- **(Introduction)** Reviewer jPp9 pointed out that most of the efficient IL algorithms we compared belong to RL-combined IL algorithms. Based on the reviewer's suggestion, we have reorganized our contributions.

- **(Related Work)** Additionally, reviewer jPp9 noted that our related work did not clarify the position of ADR in IL. Therefore, we have introduced traditional BC, the types of RL-combined IL, and mentioned the efficiency of such algorithms. We also pointed out the limitations of RL-combined IL under suboptimal rewards, followed by an introduction to ADR. Compared to the previous version, the related work in the current version has stronger logical coherence.

- **(Preliminary)** In addition, reviewer 8KXJ pointed out that DICE encompasses multiple types of distance constraints. Therefore, we have replaced the kl-divergence in the formula with divergence.

- **(Preliminary)** Finally, we have removed the content related to DICE in the preliminary section, and we have moved Definition 1 to the problem formulation.

- **(lines 464~466)** Reviewer 8KXJ pointed out that the y-axes of Figures 7 and 8 lacked information. Therefore, we added descriptions of the vertical axes for these two figures in their captions. The reason we did not choose to add the information directly on the y-axes is that these figures contain a large amount of information and are not simple line graphs or bar charts, which require separate explanations.

*Our contribution*

Our main contribution is ADR, a novel single-step supervised IL method. Unlike most modern RL-combined IL algorithms, which rely on the Bellman operator and incorporate reward shaping and Q-estimating processes, ADR operates as a single-step supervised learning paradigm, rendering it immune to the accumulated offsets resulting from suboptimal rewards. Meanwhile, ADR neither requires the addition of conservative terms nor extensive hyperparameter parameter tuning during the training process.  Meanwhile, compared to traditional single-step IL paradigms such as Behavioral Cloning (BC), ADR can achieve better performance with a limited number of demos based on adversarial density-weighted regression. Therefore, ADR combines the advantages of single-step updates while demonstrating superior performance compared to previous RL-combined IL approaches on the experimental level. Moreover, we prove that optimizing ADR’s objective is akin to approaching the demo policy, and our experimental results validate this claim, demonstrating that ADR outperforms the majority of RL-combined approaches across diverse domains.

---

### Author Response · Authors · 2024-12-02
**Dear AC, Reviewers. We discussed some concerns in our conversation with the reviewer eWnu.**

Dear AC and Reviewers,

Firstly, we would like to express our sincere gratitude to all the reviewers for taking their valuable time to write comments. We have benefited greatly from the suggestions of some of the reviewers, which have made our manuscript more comprehensive.

Meanwhile, we would like to express our gratitude to eWnu for participating in the review process. However, there are some concerns that have arisen during eWnu's comments, specifically:

- eWnu always strives to reject the paper based on what they consider to be "toy examples." Firstly, we do not object to the use of toy examples in the scientific research process. In the early stages of selecting a research topic, toy examples can be used for screening different approaches. However, our experiments have far exceeded the scope of toy examples, and additionally, the reviewer's theoretical stance on this issue is hardly tenable.

  Specifically, reviewer eWnu expresses significant concern about the case where $P^* = \hat{P}$, and cites a flawed toy example, namely, if $P^*$ and $\hat{P}$ have the same input, then $\log\frac{\hat{P}}{P^*} = 0$, leading to the phenomenon of vanishing gradients. We have reiterated our stance to the reviewer from both theoretical and experimental perspectives, emphasizing that this is unlikely to occur, and provided reasons as follows:

  1. The first is through analysis. The demo only contains one trajectory, while the suboptimal dataset has far more than that (over 1000). If the suboptimal dataset differs significantly from the demo, their densities are unlikely to be the same. Even if the suboptimal pose dataset fragements were the same as the demo, their densities would still differ due to different data distributions.

  2. The second is through experimentation. I presented experiments comparing demo to medium-expert and demo to medium. The experimental results indicate that the former outperforms the latter. If eWnu's conclusion is correct, then the performance of the two should be the same.

|medium-expert|score|medium|score|
|--|--|--|--|
|hopper-me |109.1$\pm$3.2 |hopper-m |69.0$\pm$1.1|
|halfcheetah-me| 74.3$\pm$2.1 |halfcheetah-m| 44.0$\pm$0.1|
|walker2d-me |110.1$\pm$0.2 |walker2d-m |86.3$\pm$1.7|
|Ant-me| 132.7$\pm$0.3 |Ant-m |106.6$\pm$0.5|
|Avg.| 106.6 |Avg. |76.5|
- eWnu has not made any direct review contributions to the core issues addressed in this paper, therefore, eWnu's motivation is not very clear.

- There are some thoughtless issues in eWnu's comments. For example, why use undecoded formulas as part of the review? In a public review system, shouldn't one use more professional markdown syntax? This can cause confusion for both the reviewer and the readers when reading the comments.

We are deeply grateful to AC and the other reviewers for taking your valuable time to consider our comments.

Best Regard,

Conference Authors of paper 166

---

### Meta-Review · Area_Chair_ipk8 · 2024-12-23

**Metareview:**

Thank you for your submission to ICLR. This paper presents Adversarial Density Regression (ADR), a supervised imitation learning method, that aims to mitigate limitations of existing imitation learning algorithms, such as reliance on the Bellman operator and issues resulting from out-of-distribution data.

This was a borderline submission. The reviewers agreed that the setting is well-motivated, the method avoids certain complexities of prior algorithms, and empirical results on the presented experiments look strong. However, there were still a number of unresolved concerns about the clarity of presentation in describing the relation to prior work (e.g., to DICE methods), significance of the contribution, and theoretical results. One reviewer had a particular concern with theoretical validity of the ADR loss and brought up a specific counterexample. After a thorough discussion, the reviewer was left unconvinced by the authors’ arguments (and after reading through the full discussion I am also unconvinced). There was also a substantial reformulation to the story of the paper, involving a title change and modification of content related to DICE methodology (brought up by multiple reviewers). These changes merit another round of reviews for this updated version of the paper. Based on this, and the above unresolved concerns from reviewers, I recommend this paper for rejection.

**Additional Comments On Reviewer Discussion:**

After the author rebuttal, there were a number of discussions between the authors and reviewers. This led a couple of the reviewers to increase their scores, though two reviewers had substantial remaining concerns with the paper.

---

### Decision · Program_Chairs · 2025-01-22

Reject